# Exploiting the Replay Memory Before Exploring the Environment: Enhancing Reinforcement Learning Through Empirical MDP Iteration

**Hongming Zhang[1], Chenjun Xiao[2], Chao Gao[3], Han Wang[1], Bo Xu[4], Martin Müller[1]**
[1]Department of Computing Science and Amii, University of Alberta
[2]CUHK-Shenzhen
[3]Edmonton Research Center, Huawei Canada
[4]Institute of Automation, Chinese Academy of Sciences
`hongmin2@ualberta.ca`

## Abstract

Reinforcement learning (RL) algorithms are typically based on optimizing a Markov Decision Process (MDP) using the optimal Bellman equation. Recent studies have revealed that focusing the optimization of Bellman equations solely on in-sample actions tends to result in more stable optimization, especially in the presence of function approximation. Upon on these findings, in this paper, we propose an Empirical MDP Iteration (EMIT) framework. EMIT constructs a sequence of empirical MDPs using data from the growing replay memory. For each of these empirical MDPs, it learns an estimated Q-function denoted as $\widehat{Q}$. The key strength is that by restricting the Bellman update to in-sample bootstrapping, each empirical MDP converges to a unique optimal $\widehat{Q}$ function. Furthermore, gradually expanding from the empirical MDPs to the original MDP induces a monotonic policy improvement. Instead of creating entirely new algorithms, we demonstrate that EMIT can be seamlessly integrated with existing online RL algorithms, effectively acting as a regularizer for contemporary Q-learning methods. We show this by implementing EMIT for two representative RL algorithms, DQN and TD3. Experimental results on Atari and MuJoCo benchmarks show that EMIT significantly reduces estimation errors and substantially improves the performance of both algorithms.

## 1  Introduction

Reinforcement learning (RL) has achieved remarkable success across various domains, such as games [1, 2, 3, 4, 5], robotics [6, 7, 8] and industrial applications [9, 10], by modeling them as Markov Decision Processes (MDPs). Most RL methods store transitions in a replay memory [11] and estimate an action-value function from batches of that data. They apply the Bellman optimality equation as an iterative update:

$$Q(s,a) \leftarrow r(s,a) + \gamma \max_{a'} Q(s',a'). \tag{1}$$

Such value iteration converges to the optimal action value $Q^*$ given infinite state-action visitation and updates [12, 13, 14, 15]. An optimal policy is derived by taking an action with maximum $Q$ at each state or approximated by a parameterized policy [16, 17, 18, 19]. In practice, data coverage in the replay memory is limited to a potentially small subset of the whole state-action space, especially for complex environments. Indeed, the Bellman update in Eq. (1) suffers from estimation errors due to the combination of *applying the max operator to out-of-sample actions* and *bootstrapping from a function approximator*. In the examples in Fig. 1a, the well-known *double Q-learning* method [20] cannot

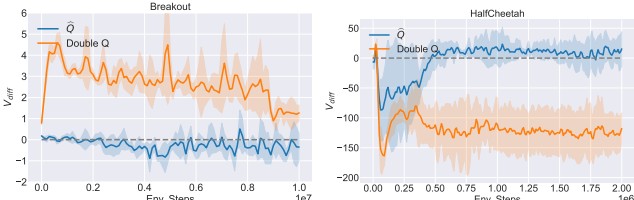
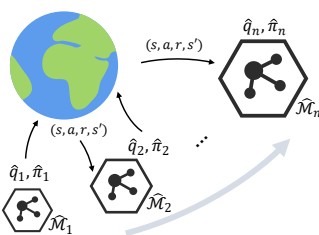

(a) Comparison of double Q technique with empirical $\widehat{Q}$. We train *Double DQN* [21] and *TD3* [17] with the double Q technique. Concurrently, we learn $\widehat{Q}$ based on Eq. (2). The graphs show the difference between the $Q$ value and discounted Monte Carlo return. Greater/less than 0 means overestimation/underestimation. The double Q technique overestimates on Breakout but underestimates on HalfCheetah. $\widehat{Q}$ yields much more accurate estimation on both tasks.

(b) In Empirical MDP Iteration (EMIT), we consider transitions existing in the replay memory as an empirical MDP $\widehat{\mathcal{M}}_i$, then solve $\widehat{\mathcal{M}}_i$ and collect more data. Repeating this process yields refined empirical MDPs $\widehat{\mathcal{M}}_1, \widehat{\mathcal{M}}_2, \cdots$, which progressively approach the original MDP $\mathcal{M}$.

eliminate the estimation error. Its effects are heavily task-specific: it overestimates on `Breakout` game but underestimates on `HalfCheetah`.

We argue that the estimation error is hard to correct for methods based on Eq. (1) that consider the entire MDP, because there are infinitely many suboptimal solutions consistent with the Bellman equation when the data is incomplete. Errors due to such incomplete data can backpropagate to transitions in the replay memory and impede the whole learning process. However, bootstrapping only from in-sample actions without querying the values of unseen actions greatly reduces the estimation error, even with function approximation. The function $\widehat{Q}$ in Fig. 1a was learned in this way.

Motivated by these observations, we advocate for solving the *empirical MDP* $\widehat{\mathcal{M}}$ which only uses transitions in a replay memory $\mathcal{D}$ collected from the environment.

$$\widehat{Q}(s,a) \leftarrow r + \gamma \max_{a':(s',a') \in \mathcal{D}} \widehat{Q}(s',a'). \tag{2}$$

There is no out-of-sample bootstrapping in Eq. (2), and the solution in a finite state-action space is unique. If $\mathcal{D}$ grows over time and gradually covers the entire MDP, then optimizing with Eq. (2) eventually leads to the solution to the original MDP. This observation instantly implies a new iterative learning procedure: alternate between *exploitation by solving an incumbent empirical MDP*, and *exploration for growing the empirical MDP with new data*. This Empirical MDP Iteration (EMIT) process is illustrated in Fig. 1b. How to grow $\widehat{\mathcal{M}}$? In principle, exploration should find missing strong actions that bring $\widehat{\mathcal{M}}$ closer to $\mathcal{M}$. Following the well-known principle of *optimism in the face of uncertainty* [22], we argue that one option is to drive the exploration for growing $\widehat{\mathcal{M}}$ by the value difference between $Q$ and $\widehat{Q}$. EMIT can be applied to any RL algorithms that can learn both $Q$ and $\widehat{Q}$ [23]. We implement and evaluate two variants based on DQN [16] and TD3 [17], for discrete action spaces and continuous control tasks respectively. Our contributions are summarized as follows:

1. We provide a thorough analysis to show why the estimation error is hard to eliminate when using the Bellman update with incomplete data, and that bootstrapping from in-sample transitions greatly reduces this error, both with and without function approximation.

2. In the tabular case, we prove that the in-sample bootstrapping guarantees a unique optimal $\widehat{Q}$ for $\widehat{\mathcal{M}}$. If the optimal trajectory for $\mathcal{M}$ is included in $\widehat{\mathcal{M}}$, then the greedy policy derived from $\widehat{Q}^*$ is also optimal for $\mathcal{M}$ following the optimal trajectory. Further, monotonic improvement in learning $\widehat{Q}$ is guaranteed with growing data coverage if $\mathcal{M}$ is deterministic. These imply that $\widehat{Q}$ can be a natural regularizer for $Q$.

3. We develop a novel framework EMIT, which can be used to enhance existing RL algorithms by iteratively solving a current empirical MDP for stable finite-time performance, and can progressively approach a solution to the original MDP. EMIT can be combined with any method that learns a $Q$ function. We add EMIT to DQN and TD3, and conduct extensive experiments on Atari [24] and MuJoCo [25] tasks. We show strong performance enhancements for both methods. Further analysis demonstrates that EMIT largely overcomes problems of estimation error.

## 2 Background

**Markov Decision Process (MDP).** Reinforcement learning (RL) [26, 27] is a paradigm of agent learning via interaction. It can be modeled as a Markov Decision Process (MDP), a 5-tuple $\mathcal{M} = (\mathcal{S}, \mathcal{A}, R, P, \gamma)$. $\mathcal{S}$ denotes the state space, $\mathcal{A}$ denotes the action space, $P(s'|s, a) : \mathcal{S} \times \mathcal{A} \times \mathcal{S} \to [0, 1]$ is the environment dynamics, $R(s, a) : \mathcal{S} \times \mathcal{A} \to \mathbb{R}$ is the reward function which is bounded, $\gamma \in (0, 1)$ is the discount factor. The goal of an RL agent is to learn an optimal policy $\pi$ that maximizes the expected discounted cumulative reward $\mathbb{E}_\pi[\sum_{t=0}^\infty \gamma^t r_t]$.

**Value-Based Methods.** Q-learning is a classic algorithm based on the Bellman optimality equation $Q^*(s, a) = \mathbb{E}[r + \gamma \max_{a'} Q^*(s', a')]$ [28]. An optimal policy takes an action with maximum $Q$ at each state. DQN [16] scales up from tabular $Q$-learning by using deep neural networks and experience replay [11]. Actor-critic methods [6] such as SAC [18] and TD3 [17] learn a parameterized policy, which is suitable for continuous action spaces. The policy is updated by gradient ascent of a $Q$ network, $\nabla_\theta \frac{1}{|\mathcal{B}|} \sum_{s \in \mathcal{B}} Q_\phi(s, a_\theta), a_\theta \sim \pi_\theta(\cdot|s)$, where $\mathcal{B}$ is a batch of samples.

**In-Sample Bellman Update.** The in-sample constraint in Eq. (2) avoids bootstrapping from unseen actions when estimating target values. Several recent offline RL [29] methods use Eq. (2) to approximate an optimal action value function. For example, implicit $Q$-learning (IQL) [30] uses expectile regression to learn the optimal $Q$ function without ever querying the values of unseen actions. Its learning goal is to minimize the expectile regression objective:

$$L(\theta) = \mathbb{E}_{(s,a,s',a') \sim \mathcal{D}}[L_2^\tau(r(s, a) + \gamma Q_{\bar{\theta}}(s', a') - Q_\theta(s, a))], \tag{3}$$

where $L_2^\tau(u) = |\tau - \mathbb{1}(u < 0)|u^2$ is a weighted mean squared error loss, and $Q_{\bar{\theta}}$ is the target network. This asymmetric loss function defines the entire spectrum of methods between SARSA ($\tau = 0.5$) and $Q$-learning ($\tau \to 1$). Eq. (2) can be approximated using Eq. (3) with $\tau \approx 1$. Other methods like In-Sample Actor-Critic (InAC) [31] and Extreme Q-learning (XQL) [32] are also based on Eq. (2), and EMIT is straightforward to implement based on these offline methods.

## 3 Empirical MDP Iteration

A replay memory $\mathcal{D}$ is maintained to store transitions $(s, a, r, s')$. In most cases, the original MDP $\mathcal{M}$ is too large, and $\mathcal{D}$ contains only a small subset of all transitions in $\mathcal{M}$. Due to the incomplete data, infinitely many other MDPs can be consistent with $\mathcal{M}$ on transitions in $\mathcal{D}$. For ease of notation, we denote $(s, a) \in \mathcal{D}$ if $\exists (s, a, r, s') \in \mathcal{D}$ and $s \in \mathcal{D}$ if $\exists (s, a, r, s') \in \mathcal{D}$ or $\exists (\tilde{s}, a, r, s) \in \mathcal{D}$. A visit count $N(s, a, s')$ is defined as the number of times $(s, a, s')$ appears in $\mathcal{D}$. We define the empirical MDP $\widehat{\mathcal{M}}$ to be the lowest reward MDP that uses all data in $\mathcal{D}$.

**Definition 3.1** (Empirical MDP). Given a dataset $\mathcal{D}$ from MDP $\mathcal{M}$, the empirical MDP $\widehat{\mathcal{M}} := (\widehat{\mathcal{S}}, \widehat{\mathcal{A}}, \widehat{R}, \widehat{P}, \gamma)$, has state space $\widehat{\mathcal{S}} = \{s|s \in \mathcal{D}\}$, and action space $\widehat{\mathcal{A}} = \{a|(s, a) \in \mathcal{D}\}$, with reward function $\widehat{R}(s, a) = R(s, a)$ if $(s, a) \in \mathcal{D}$, and $\widehat{R}(s, a) = -\infty$ otherwise. $\widehat{P}(s'|s, a) = \frac{N(s,a,s')}{\sum_{s'} N(s,a,s')}$ is the empirical transition dynamics based on visit counts in $\mathcal{D}$, and $\gamma \in (0, 1)$ is the discount factor.

The empirical MDP $\widehat{\mathcal{M}}$ contains all information we can obtain from $\mathcal{D}$. If a (state, action) pair is not in the data, its reward is set to $-\infty$ and its transition probabilities are zero.

We analyze two update rules in the tabular case with finite $\mathcal{S} \times \mathcal{A}$, the *Bellman update* and the *in-sample Bellman update* given in Eqs. (1) and (2). Both update rules try to minimize the Bellman residual [33], which is a surrogate objective to minimize the action value error and approximate the optimal value. Eq. (1) takes the maximum over all actions, while Eq. (2) only uses in-sample actions for a state. Let $Q^*$ and $\widehat{Q}^*$ be the optimal action values for $\mathcal{M}$ and $\widehat{\mathcal{M}}$ respectively. Then:

**Proposition 3.2.** *If the data coverage $\mathcal{D}$ is incomplete, then the Bellman update Eq. (1) neither guarantees convergence to the optimal value $Q^*$ for the original MDP $\mathcal{M}$ nor to the optimal value $\widehat{Q}^*$ for the empirical MDP $\widehat{\mathcal{M}}$, even in the limit of infinite updates.*

The proof is straightforward. Given that $\mathcal{D}$ is incomplete, there exist transitions in $\mathcal{M}$ that are not present in $\widehat{\mathcal{M}}$. These absent transitions can be assigned any initial values, and may bootstrap to in-sample transitions, thereby influencing the convergence process. This indicates that the Bellman

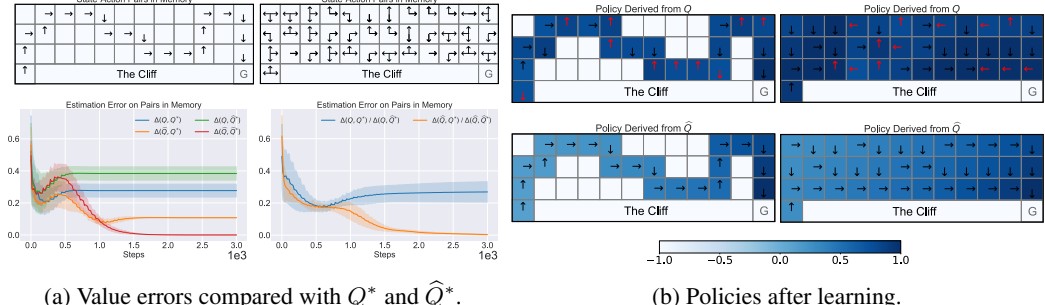

(a) Value errors compared with $Q^*$ and $\widehat{Q}^*$.     (b) Policies after learning.

Figure 2: (a) illustrates two cases in the CliffWalk task: memory $\mathcal{D}$ only contains a sub-optimal trajectory and misses many transitions (left); $\mathcal{D}$ contains all optimal state-action pairs but misses some sub-optimal actions (right). The curves show estimation errors of $Q$ and $\widehat{Q}$ learned with Eqs. (1) and (2) compared with $Q^*$ and $\widehat{Q}^*$. (b) presents the greedy policies after learning. Red arrows indicate incorrect actions neither follow $\arg\max Q^*$ nor $\arg\max \widehat{Q}^*$. The blue shading signifies accurately estimated $\widehat{Q}$ and overestimated $Q$ values.

update is heavily influenced by the values of unseen state-action pairs. A simple example is provided in Appendix A Fig. 7. In contrast, the in-sample Bellman update is not affected by unseen transitions, and thus converges to a unique optimal value $\widehat{Q}^*$ for $\widehat{\mathcal{M}}$.

**Proposition 3.3.** *The in-sample Bellman update Eq. (2) uniquely converges to the optimal value $\widehat{Q}^*$ for the empirical MDP $\widehat{\mathcal{M}}$ in the limit of infinitely many updates. Furthermore, if the optimal trajectory for $\mathcal{M}$ is included in $\widehat{\mathcal{M}}$, then the greedy policy derived from $\widehat{Q}^*$ is also optimal for $\mathcal{M}$ following the optimal trajectory.*

We defer the proof to Appendix A, and provide a toy example on CliffWalk [26] in Fig. 2 to illustrate the difference between the two update rules. CliffWalk is a simple navigation task. The goal is to reach the state G at the bottom right starting from the bottom left. The reward of reaching G is 1. Dropping into the cliff gives reward -1, and all other rewards are 0. We present two cases: (1) $\mathcal{D}$ contains a sub-optimal trajectory and misses many state-action pairs (Fig. 2a top left), and (2) $\mathcal{D}$ contains an optimal action at each state but misses some sub-optimal actions (Fig. 2a top right).

We parameterize $Q$ and $\widehat{Q}$ with neural networks and sample batches of data to update them with gradient descent following Eqs. (1) and (2). The learning curves in Fig. 2a shows value errors comparing with the optimal values. $\Delta(Q_1, Q_2)$ denotes the averaged absolute error on state-action pairs in $\mathcal{D}$ between $Q_1$ and $Q_2$. $\Delta(\widehat{Q}, \widehat{Q}^*)$ converges to 0, indicates that $\widehat{Q}$ converges to $\widehat{Q}^*$. However, $Q$ neither converges to $Q^*$ nor $\widehat{Q}^*$. In particular in the second case, all optimal actions are included in $\mathcal{D}$. $\widehat{Q}$ converges to global optimum while $Q$ still fails due to missing actions. This highlights the heavy influence of missing transitions on the Bellman update. Fig. 2b further presents the greedy policies derived from $Q$ and $\widehat{Q}$ after learning. Red arrows present actions that neither follow $\arg\max Q^*$ nor $\arg\max \widehat{Q}^*$. The blue shading indicates the action values. $Q$ overestimates on most state-action pairs (top row of Fig. 2b). The policy derived from $Q$ cannot learn the (current) optimal actions, while $\arg\max \widehat{Q}$ represents a best possible policy.

We solve each empirical MDP $\widehat{\mathcal{M}}_i$, defined by the current memory $\mathcal{D}_i$, and add data to progressively approach the original MDP $\mathcal{M}$. If $\mathcal{M}$ is deterministic, a monotonic improvement of the learning is guaranteed by alternating between learning a currently best possible policy and collecting more data.

**Proposition 3.4.** *Assume $\mathcal{M}$ has deterministic transitions. If $\{\mathcal{D}_i\}$ are datasets collected from $\mathcal{M}$ with $\mathcal{D}_1 \subset \mathcal{D}_2 \subset \cdots \subset \mathcal{D}_n$ and corresponding empirical MDPs $\{\widehat{\mathcal{M}}_i\}$, then $\widehat{Q}_1^* \le \widehat{Q}_2^* \le \cdots \le \widehat{Q}_n^*$.*

This result is intuitive. With more transitions in the replay memory, we take the maximum over a wider range of actions, leading to a better solution closer to $Q^*$. If the empirical MDP contains all transitions, then $\widehat{\mathcal{M}} = \mathcal{M}$ and $\widehat{Q}^* = Q^*$. The proof details are provided in Appendix A.

The in-sample Bellman update provides a learning path to improve a policy. The guarantee of the convergence to a unique optimal value on existing transitions is a key property, which makes the in-sample Bellman update a more robust choice for learning in the presence of missing transitions.

More importantly, if the globally optimal trajectory has been explored and included in the memory, then the in-sample Bellman update can learn an optimal policy catching up with the optimal trajectory even when some other transitions are missing. In contrast, the Bellman update is strongly affected by such missing transitions, and cannot guarantee to learn even from optimal trajectories.

### 3.1 Enhancing Online RL Algorithms with EMIT

Two major questions in EMIT are how to learn $\widehat{Q}$ from $\widehat{\mathcal{M}}$ and how to grow $\widehat{\mathcal{M}}$ using out-of-the-box strong reinforcement learning algorithms. To take advantage of an existing value-based learning algorithm, we utilize $\widehat{Q}$ to modify $Q$ learning in two places, based on the value difference. First, given a sampled batch of data, we modify the standard mean squared error (MSE) loss function of $Q$ by adding a MSE loss of $\widehat{Q}$ as a regularizer:

$$\mathcal{L} = MSE(Q_\theta, Q_{\text{target}}) + \alpha MSE(Q_\theta, \widehat{Q}), \tag{4}$$

Here, $Q_{\text{target}} = r(s,a) + \gamma \max_{a'} Q(s', a')$, and $\alpha$ is a parameter controlling the level of regularization. Second, we add an exploration bonus to the algorithm's behavior policy. The rationale is that since we regularize $Q$ to approximate $\widehat{Q}$ on existing transitions in the replay memory, the values of $Q(s,a)$ and $\widehat{Q}(s,a)$ should be closer at known than at unseen state-action pairs. Therefore, we provide more incentive for the agent to explore those dissimilar states. Define the absolute difference between $Q$ and $\widehat{Q}$ at $(s,a)$ when interacting with the environment by:

$$\delta(s,a) = |Q(s,a) - \widehat{Q}(s,a)|. \tag{5}$$

For algorithms in discrete action spaces such as DQN, we can add an exploration bonus $\delta(s,a)$ to the greedy part of an $\epsilon$-greedy policy: with probability $\epsilon$ it chooses an random action as usual, otherwise it selects an action that is greedy w.r.t $Q(s,a) + \delta(s,a)$:

$$\pi = \begin{cases} \text{random action,} & p = \epsilon \\ \arg\max_a(Q(s,a) + \delta(s,a)), & p = 1 - \epsilon \end{cases} \tag{6}$$

Eq. (6) adds a targeted exploration mechanism to $\epsilon$-greedy.

For algorithms in continuous action spaces such as TD3, where the policy is assumed to be a unimodal Gaussian distribution, we use $\delta$ as a state-action dependent standard deviation and sample actions according to:

$$a \leftarrow \pi(s) + \epsilon, \quad \epsilon \sim \text{clip}(\mathcal{N}(0, \delta(s,a)), -c, c), \tag{7}$$

The added noise is clipped to keep the target close to the original action [17].

Algorithm 1 shows pseudo-code for incorporating EMIT into a $Q$ learning algorithm `Alg`. Blue text highlights the components added or modified by EMIT. `EMIT` maintains $\widehat{Q}$ and incorporates it into the algorithm's action selection `Alg.act()` and $Q$ update `Alg.update()`. Any method based on Eq. (2) can be used to learn $\widehat{Q}$. In our experiments, we apply implicit $Q$-learning (IQL) [30]. We learn $Q$ using DQN [16] and TD3 [17] for discrete and continuous environments respectively. Clearly, Algorithm 1 should be more appropriate be viewed as a new framework, rather than a fixed algorithm.

---

**Algorithm 1** Empirical MDP Iteration for Enhancing a $Q$ Learning Algorithm `Alg` (EMIT-Alg)

---

1: `Alg.Initialize` the replay memory $\mathcal{D}$ and action value network $Q_\theta$
2: `EMIT.Initialize` $\widehat{Q}_{\hat\theta}$
3: `Alg.Initialize` the environment $s_0 \leftarrow Env$
4: **for** environment step $t = 0$ **to** $T$ **do**
5:     `Alg.Select` an action $a_t = $ `Alg.act`$(s_t, \widehat{Q}_{\hat\theta})$ as Eq. (6) or (7) {$\widehat{Q}$ guided exploration}
6:     `Alg.Execute` $a_t$ in $Env$ and get $r_t, s_{t+1}$
7:     `Alg.Store` transition $(s_t, a_t, r_t, s_{t+1})$ in $\mathcal{D}$
8:     `Alg.Sample` random minibatch of transitions $\mathcal{B}$ from $\mathcal{D}$
9:     `EMIT.update`$(\widehat{Q}_{\hat\theta}, \mathcal{B})$ w.r.t MSE loss derived by Eq.(2) {Learning of $\widehat{Q}$}
10:     `Alg.update`$(Q_\theta, \mathcal{B}, \widehat{Q}_{\hat\theta})$ as Eq.(4) {$\widehat{Q}$ regularized learning for $Q$}
11: **end for**

---

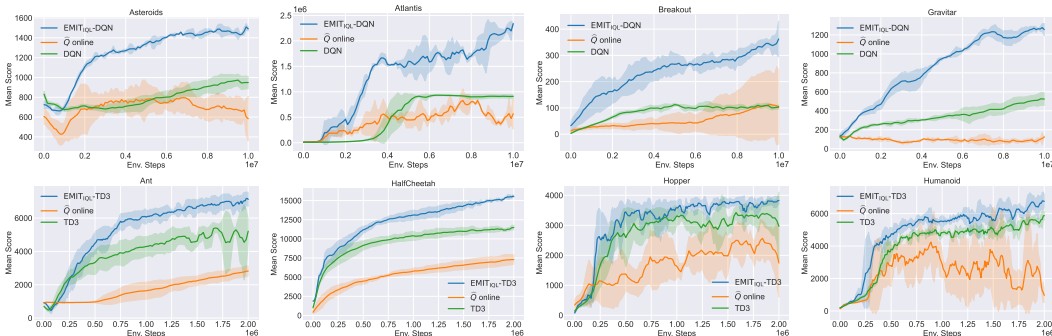

Figure 3: Performance on Atari (first row) and MuJoCo (second row) tasks. The learning curves show mean scores and standard deviations over five runs. EMIT consistently boosts performance across diverse tasks.

## 4 Experiments

In this section, we present experimental results to validate the effectiveness of our proposed method. We applied EMIT to both DQN [16] and TD3 [17], covering both discrete and continuous action spaces. Our results demonstrate a significant performance improvement when EMIT is applied to these current DRL methods. Ablation studies underscore the importance of each component of our method in contributing to this improvement. Further analysis provides insights into why EMIT works. Notably, the in-sample Bellman update Eq. (2) is capable of learning a superior policy compared to the Bellman update Eq. (1). Regularization of $Q$ by $\widehat{Q}$ yields a more accurate estimate, and the estimation error in $Q$ decreases. Additionally, the observed decrease in policy churn [34] may represent another potential benefit of our method, worth further investigation.

### 4.1 Setup

**Environments.** We evaluate EMIT on Atari [24] and MuJoCo [25] tasks based on OpenAI Gym interface [35]. Atari provides a diverse set of video games, making it an ideal platform for evaluating the general competency of AI agents. The input for these games is image-based, and the dynamics are nearly deterministic. MuJoCo offers a set of continuous control tasks, all of which have a standardized structure and interpretable rewards. These tasks are designed to facilitate research and development in the field of robotics, where the need for fast and accurate simulation is paramount. The dynamics in MuJoCo are deterministic and feature high-dimensional state features. More details about these environments can be found in Appendix B.

**Baselines and Implementation Details.** We benchmark EMIT against several established algorithms. For Atari games, we compare our method with DQN [16], C51 [36], IQN [37], and Rainbow [38]. Each run involves 10 million interaction steps. Performance is evaluated by executing 30 episodes after every 100k environmental steps. For MuJoCo tasks, our method is compared with SAC [18, 39], TD3 [17], XQL [32], TRPO [40], and PPO [41]. Each run involves 2 million interaction steps. Performance is evaluated by executing 10 episodes after every 10k environmental steps. We use the same network as in the original papers for each algorithm. We search the learning rates for baselines among {1e-3,3e-4,1e-4} and report the best performance. Each experiment is run with 5 different random seeds. For our method, we set the learning rate the same as the backbone algorithm, and search for the best regularization parameter $\alpha \in \{0.05, 0.1, 0.5\}$. All other common parameters are set as detailed in Appendix B Tables 1 and 2. Further details can be found in Appendices B.2 and B.3.

### 4.2 Performance Enhancement of EMIT for DQN and TD3

To demonstrate the effectiveness of EMIT, we initially compare it with its backbone algorithms. Fig. 3 depicts the learning curves for Atari games (first row) and MuJoCo tasks (second row). The solid line represents the mean score, while the shaded area indicates the standard deviation. The $\widehat{Q}$ **online** result refers to an IQL agent run online with the learning rule Eq. (2). This agent attempts to solve the empirical MDP but lacks the exploration mechanism that augments the MDP. DQN and TD3 are agents that follow the learning rule Eq. (1). They have exploration mechanisms such as

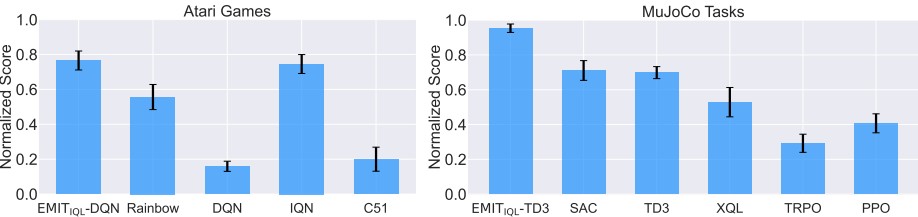

Figure 4: Average performance on 20 Atari games and 8 MuJoCo tasks, with each method's score normalized between 0 and 1. EMIT achieves the best average performance across these tasks.

$\epsilon$-greedy and random noise. They differentiate from the EMIT counterparts in that they do not learn from the empirical MDP, hence they lack the regularization and exploration bonus derived from $\widehat{Q}$. Our EMIT methods learn both $Q$ and $\widehat{Q}$. As depicted in Fig. 3, they achieve clear improvements on all tasks compared to other methods that solely learn $Q$ or $\widehat{Q}$. EMIT can effectively augment and boost these existing RL algorithms. It provides a principled framework for seamlessly integrating the strengths of traditional learning based on the $Q$ function and the more recent advancements in in-sample learning for $\widehat{Q}$. Additional results can be found in Appendix C Figs. 23 and 24.

## 4.3 Comparison with Other Baselines

We extend our comparison of EMIT with other strong baselines. For a clearer presentation, we display normalized scores. For each task, we define $R_{\max}$ and $R_{\min}$ as the maximum and minimum cumulative reward across all methods, respectively. The score for each method is then normalized using the formula $(R - R_{\min})/(R_{\max} - R_{\min})$.

The overall performance on various tasks is presented in Fig. 4. EMIT achieves the best average performance across these tasks. Specifically, EMIT either outperforms or is on par with (within a 10% difference) the best baselines on 19 out of 28 tasks. Details are given in Appendix C.2 Fig. 17. Notably, on MuJoCo tasks, EMIT significantly surpasses other baselines. These results corroborate our intuition that Empirical MDP Iteration is a more effective approach than focusing on the original MDP from the outset. This strategy circumvents bootstrapping error, thereby enhancing sample efficiency. While the Double Q technique mitigates overestimation by employing the target network to evaluate actions, the target network may still overestimate or underestimate certain state-action pairs. The distributional perspective of value estimation fosters more stable and risk-aware behavior. However, estimation error persists due to the maximization bootstrapping from out-of-sample actions. In contrast, EMIT seeks to simplify the problem, addressing it incrementally and progressively approximating the original problem. This approach can provide accurate target values for each empirical MDP, leading to consistent improvement. In summary, EMIT reduces estimation error and provides accurate targets, thereby making the learning process more efficient. Detailed learning curves can be found in Appendix C.2.

## 4.4 How the In-Sample Bellman Update Benefits the Learning

In Section 3, we analyze beneficial properties of the in-sample Bellman update. In this section, we provide comprehensive empirical evidence for them.

We first demonstrate that the in-sample Bellman update is well-suited for passive learning [42, 29], without the need for active interaction with the environment. Since the policy under which the data is collected is not a concern, this allows greater flexibility in designing exploration strategies. We execute DQN and TD3 (based on the Bellman update in Eq. (1)) on Breakout and HalfCheetah to learn a policy $\pi$. Concurrently, we learn a policy $\hat{\pi}$ using another IQL agent with in-sample Bellman updates Eq. (2). Policy $\pi$ interacts with the environment and collects data into a replay memory. Both $\pi$ and $\hat{\pi}$ learn from this same dataset. Fig. 5a illustrates the performance of $\pi$ and $\hat{\pi}$. Interestingly, $\hat{\pi}$ performs comparably, if not better than $\pi$. Despite $\hat{\pi}$ being learned without taking actions in the environment, we observe no performance degradation attributable to passive learning. This is somewhat counterintuitive, as active learning is generally considered superior [43].

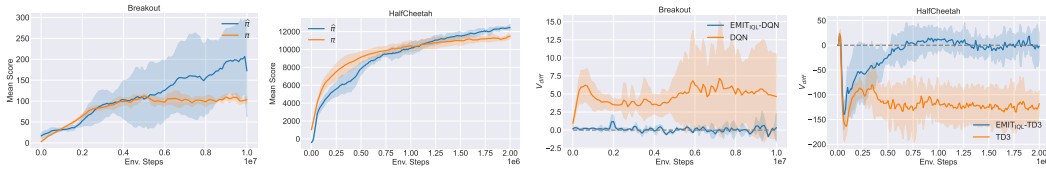

(a) Policy performance.  (b) Estimation error of $Q$ after regularization.

Figure 5: (a) We run DQN and TD3 on Breakout and HalfCheetah, and concurrently learn action value $\widehat{Q}$ with the collected data. The curves show the performance of policies $\pi$ and $\hat{\pi}$, derived from $Q$ and $\hat{Q}$ respectively. Remarkably, $\hat{\pi}$, learned without active environment interaction, matches or even surpasses $\pi$'s performance. (b) EMIT helps reduce the error in $Q$ and learns almost accurate value estimation during the learning process.

We conclude that the in-sample Bellman update can learn an effective policy from given data without the need for active interaction with the environment, and that the active policy $\pi$ does not fully utilize the collected data and fails to identify the optimal policy due to bootstrapping errors. This underscores that bootstrapping from out-of-sample transitions hampers sample-efficient learning. Similar results on additional environments can be found in Appendix C.3 Figs. 18 and 19.

Next, we demonstrate that the regularization of $Q$ with $\widehat{Q}$ significantly diminishes the estimation error in $Q$. At each evaluation, we obtain the true discounted Monte Carlo return, denoted as $Q_{\text{True}}$, by executing the current policy for 30 episodes and calculating the actual discounted cumulative rewards, $Q_{\text{True}}(s_t, a_t) = \sum_{t=t}^{T} \gamma^t r_t$. We then compute the average estimation difference between the learned $Q$ and the ground truth $Q_{\text{True}}$,

$$V_{\text{diff}}(Q, Q_{\text{True}}) = \frac{1}{|\mathcal{T}|} \sum_{(s,a) \in \mathcal{T}} \left( Q(s,a) - Q_{\text{True}}(s,a) \right), \tag{8}$$

Here, $\mathcal{T}$ represents the set of evaluation trajectories, and $|\mathcal{T}|$ denotes the total number of state-action pairs in $\mathcal{T}$. We calculate $V_{\text{diff}}$ for our method and compare it with DQN and TD3 on Breakout and HalfCheetah[1]. As shown in Fig. 5b, our method maintains the difference near zero, indicating that the estimation is remarkably close to the true value, with no significant overestimation or underestimation. Similar results for other environments can be found in Appendix C.3.

Another potential factor contributing to the performance improvement is the reduction of policy churn, the rapid change of the greedy policy in value-based reinforcement learning, primarily induced by high-variance updates in deep learning. Much of this policy change could be unnecessary, particularly as learning converges [34]. Measuring policy change in a continuous action space is not straightforward. Therefore, we assess the change on Breakout and Pong, which have discrete action spaces. The results are shown in Fig. 6a, with additional results in Fig. 22 in Appendix C.4. Our findings consistently indicate that our method reduces policy change compared to DQN. However, policy churn is not entirely detrimental, as it can serve as a potentially beneficial form of implicit exploration. The optimal frequency of change that could enhance learning remains unclear and deserves further investigation.

## 4.5 Effect of Loss Regularization and Exploration Bonus

Our algorithm design comprises two key components: the regularization term, determined by the parameter $\alpha$, and the exploration term, designed based on the value difference $\delta(s, a)$. The impact of each component on the Breakout and HalfCheetah environments is further shown in Figure 6b. The label **w/o reg term** signifies the absence of regularization for the function $Q$, i.e., $\alpha = 0$. **w/o explore term** indicates the use of existing exploration methods, such as $\epsilon$-greedy or random noise, without the addition of our exploration mechanism. **w/o both** refers to the backbone methods devoid of two enhancements. **full method** incorporates both the regularization and exploration terms. Figure 6b reveals that both components independently contribute to learning. We see a more significant performance decline without the regularization term. This suggests that, by adhering to the paradigm of empirical MDP iteration, the current exploration mechanism can already yield substantial improvements. The introduction of an advanced exploration mechanism can further enhance performance. Similar results for more environments can be found in Appendix C.5.

---

[1]Unlike Fig. 1a that measures the error of $\widehat{Q}$, here we measure the estimation error of $Q$ after regularization.

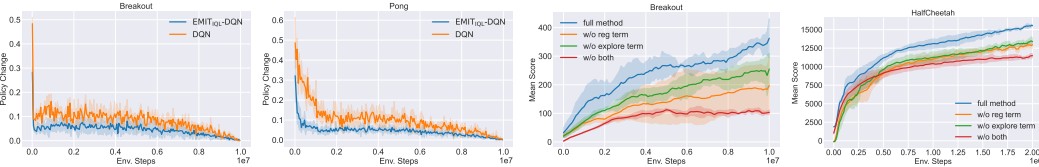

(a) The policy change over the learning.    (b) The contribution of each component of EMIT.

Figure 6: (a) EMIT helps reduce unnecessary policy change comparing with DQN, potentially contributing to the enhanced performance. (b) Both the regularization and the exploration mechanism benefit the online learning. The regularization term exerts a greater impact than the exploration mechanism.

# 5   Related Work

Accurate value estimation is crucial for RL algorithms. The use of erroneous bootstrapping targets can degrade the $Q$ function, resulting in suboptimal performance [44, 45, 46, 47]. Overestimation can occur when bootstrapping error is combined with maximizing over action values [20]. Double DQN [21] mitigates this overestimation by computing the maximum from two $Q$ functions. Maxmin Q-learning [48] learns multiple Q functions to strike a balance between overestimation and underestimation. It requires meticulous hyper-parameter fine-tuning, which can be computationally intensive. CEER [49] constructs a graph to merge similar states and computes an additional conservative $Q$ estimation akin to solving Eq. (2). However, CEER is effective only when the constructed graph is dense. C51 [36] and IQN [37] are distributional RL methods that are generally more stable and risk-aware than DQN. Nonetheless, estimation error remains due to the maximization bootstrapping from out-of-sample actions. Rainbow [38] integrates six enhancements to the DQN algorithm for improved empirical performance: double Q-learning [21], prioritized replay [50], dueling networks [51], multi-step learning [52], distributional RL [36], and noisy nets [53].

Offline reinforcement learning [29] tries to fully use existing data without additional online data collection. They aim to extract best possible policy from the existing dataset. While earlier methods focused on constraining the distance between the learned policy and the behavior policy to avoid distributional shift caused by taking actions outside of the behavior distribution [54, 55, 56], recent studies found that selecting actions within the support of the dataset during training, similar to Eq. (2), is more effective [57, 30, 58, 31]. Implicit $Q$-Learning (IQL) [30] estimates the value of the best available action at a given state with expectile regression, without ever directly querying the $Q$ function for unseen actions. In-Sample Actor-Critic (InAC) [31] approximates an in-sample softmax using only actions in the dataset. Extreme Q-learning (XQL) [32] models the maximal value using Extreme Value Theory (EVT) and avoids computing Q-values using out-of-sample actions. In EMIT, we provide a framework for easy integration of any of these methods to solve Eq. (2).

# 6   Discussion and Limitations

We study the application of the Bellman equation as a learning objective in scenarios with incomplete data. Bootstrapping from in-sample transitions with Eq. (2) theoretically leads to a unique solution and significantly reduces the estimation error in practice, even with function approximation. We introduce a novel learning paradigm, termed Empirical MDP Iteration (EMIT). Unlike previous methods that solely focus on solving the entire original MDP, we propose a regularization approach for learning by solving a series of empirical MDPs using only the transitions present in the data. EMIT provides an iterative learning pathway that uniquely solves each empirical MDP and incrementally approaches the original MDP through new data collection. We instantiate EMIT with the $Q$-learning algorithm DQN and the actor-critic algorithm TD3. Results demonstrate a substantial improvement in performance in applications of both video games and continuous control tasks. A shortcoming is that since the policy derived from $\widehat{Q}$ lacks an exploration mechanism, we learn two $Q$ functions in our algorithm design and the wall-clock time would be double. One future work could be designing a learning process that is directly based on in-sample Bellman update to avoid out-of-sample bootstrapping and also taking the exploration mechanism into consideration.

## Acknowledgements

Zhang and Müller acknowledge support from NSERC, the Natural Sciences and Engineering Research Council of Canada, UAHJIC, the Digital Research Alliance of Canada, and the Canada CIFAR AI Chair program. Wang acknowledges support from the Digital Research Alliance of Canada. Xiao acknowledges support from the National Natural Science Foundation of China (62406271).

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

# A Proof of Propositions in Section 3

In this section, we analyze the property of the two update rules in Eqs. (1) and (2) for finite state-action space $\mathcal{S} \times \mathcal{A}$ discussed in Section 3.

**Proposition 3.2.** *If the data coverage $\mathcal{D}$ is incomplete, then the Bellman update Eq. (1) neither guarantees convergence to the optimal value $Q^*$ for the original MDP $\mathcal{M}$ nor to the optimal value $\widehat{Q}^*$ for the empirical MDP $\widehat{\mathcal{M}}$, even in the limit of infinite updates.*

We illustrate the property using an example in Fig. 7. The original MDP comprises two states $\{s_1, s_2\}$ and two actions $\{a_1, a_2\}$. The memory $\mathcal{D}$ includes transitions $\{(s_1, a_1), (s_1, a_2), (s_2, a_1)\}$, but lacks $(s_2, a_2)$. Assuming a reward $r(s, a) = 0$ for all transitions for simplicity. In this case, the optimal action value for all state-action pairs in $\mathcal{M}$ and $\widehat{\mathcal{M}}$ is 0, i.e., $Q^*(s, a) = \widehat{Q}^*(s, a) = 0, a \in \{a_1, a_2\}, s \in \{s_1, s_2\}$.

However, the reward for the missing transition can take any value, leading to an infinite number of MDPs $\mathcal{M}_i \neq \mathcal{M}$, where $i \in \mathcal{I}$ denotes different $r(s_2, a_2)$ values. For the Bellman update Eq. (1), any action-value function can only converge to one of these $\mathcal{M}_i$, depending on the initialization. The action value at state $s_1$ may be determined by the value of the missing transition $(s_2, a_2)$. For instance, if $Q(s_2, a_2)$ is initialized with any positive value, Eq. (1) converges to $Q(s_1, a) = \gamma Q(s_2, a_2) > 0$, failing to approximate the optimal value 0 of either $Q^*$ or $\widehat{Q}^*$. This suggests that the Bellman update is significantly affected by out-of-sample state-action pairs, and errors propagate back to in-sample transitions. A more formal proof is available in related work [59] Corollary 1.

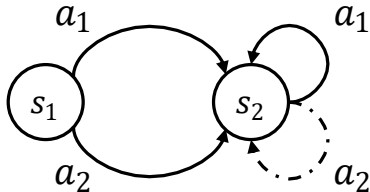

Figure 7: A toy MDP with two states and two actions. $(s_2, a_2)$ is missed in the replay memory.

**Proposition 3.3.** *The in-sample Bellman update Eq. (2) uniquely converges to the optimal value $\widehat{Q}^*$ for the empirical MDP $\widehat{\mathcal{M}}$ in the limit of infinitely many updates. Furthermore, if the optimal trajectory for $\mathcal{M}$ is included in $\widehat{\mathcal{M}}$, then the greedy policy derived from $\widehat{Q}^*$ is also optimal for $\mathcal{M}$ following the optimal trajectory.*

We first present the established results for the Bellman update in Eq. (1), and then provide a three-step proof for Proposition 3.3.

The Bellman (optimality) operator $\mathcal{B}$ for Eq. (1) is defined as:

$$(\mathcal{B}Q)(s, a) = \sum_{s' \in \mathcal{S}} P(s'|s, a)[r + \gamma \max_{a'} Q(s', a')]. \tag{9}$$

Previous works have demonstrated that the operator $\mathcal{B}$ is a $\gamma$-contraction with respect to the supremum norm:

$$\|\mathcal{B}Q_1 - \mathcal{B}Q_2\|_\infty \leq \gamma \|Q_1 - Q_2\|_\infty, \tag{10}$$

where the supremum norm $\|v\|_\infty = \max_{1 \leq i \leq d} |v_i|$, and $d$ is the dimension of vector $v$. According to Banach's fixed-point theorem [60], $Q$ converges to optimal action value $Q^*$ if we consecutively apply operator $\mathcal{B}$ to $Q$, $\lim_{n \to \infty} (\mathcal{B})^n Q = Q^*$. Furthermore, the update rule in Eq. (1), i.e. $Q$-learning, is a sampling version that applies the $\gamma$-contraction operator $\mathcal{B}$ to $Q$. It can be viewed as a random process and will converge to $Q^*$, $\lim_{t \to \infty} Q_t = Q^*$, under certain mild conditions [61, 62, 63, 64].

In a similar vein, we define the empirical Bellman (optimality) operator $\widehat{\mathcal{B}}$ for Eq. (2):

$$(\widehat{\mathcal{B}}\widehat{Q})(s, a) = \sum_{s' \in \mathcal{S}} \widehat{P}(s'|s, a)[r + \gamma \max_{a':(s',a') \in \mathcal{D}} \widehat{Q}(s', a')]. \tag{11}$$

We prove Proposition 3.3 in three steps. Firstly, we demonstrate that $\widehat{\mathcal{B}}$ is a $\gamma$-contraction operator under supremum norm, and thus converges to the optimal action value $\widehat{Q}^*$, $\lim_{n\to\infty}(\mathcal{B})^n\widehat{Q} = \widehat{Q}^*$. Next, we show that the sampling-based update rule in Eq. (2) converges to $\widehat{Q}^*$, $\lim_{t\to\infty}\widehat{Q}_t = \widehat{Q}^*$. Finally, we illustrate the greedy policy derived from $\widehat{Q}^*$ is optimal for $\mathcal{M}$ if the optimal trajectory for $\mathcal{M}$ is included in $\widehat{\mathcal{M}}$.

**Lemma A.1.** *The operator $\widehat{\mathcal{B}}$ defined in Eq. (11) is a $\gamma$-contraction operator under supremum norm,*

$$\|\widehat{\mathcal{B}}\widehat{Q}_1 - \widehat{\mathcal{B}}\widehat{Q}_2\|_\infty \leq \gamma\|\widehat{Q}_1 - \widehat{Q}_2\|_\infty. \tag{12}$$

*Proof.* We rewrite $\|\widehat{\mathcal{B}}\widehat{Q}_1 - \widehat{\mathcal{B}}\widehat{Q}_2\|_\infty$ as follows:

$$
\begin{aligned}
&\|\widehat{\mathcal{B}}\widehat{Q}_1 - \widehat{\mathcal{B}}\widehat{Q}_2\|_\infty \\
&= \max_{s,a}\Big| \sum_{s'\in\mathcal{S}} \widehat{P}(s'|s,a)[r + \gamma \max_{a_1':(s',a_1')\in\mathcal{D}} \widehat{Q}_1(s',a_1')] - \widehat{P}(s'|s,a)[r + \gamma \max_{a_2':(s',a_2')\in\mathcal{D}} \widehat{Q}_2(s',a_2')]\Big| \\
&= \max_{s,a}\gamma\Big| \sum_{s'\in\mathcal{S}} \widehat{P}(s'|s,a)[\max_{a_1':(s',a_1')\in\mathcal{D}} \widehat{Q}_1(s',a_1') - \max_{a_2':(s',a_2')\in\mathcal{D}} \widehat{Q}_2(s',a_2')]\Big| \\
&\leq \max_{s,a}\gamma \sum_{s'\in\mathcal{S}} \widehat{P}(s'|s,a)\Big|\max_{a_1':(s',a_1')\in\mathcal{D}} \widehat{Q}_1(s',a_1') - \max_{a_2':(s',a_2')\in\mathcal{D}} \widehat{Q}_2(s',a_2')\Big| \\
&\leq \max_{s,a}\gamma \sum_{s'\in\mathcal{S}} \widehat{P}(s'|s,a) \max_{\tilde{a}:(s',\tilde{a})\in\mathcal{D}}\Big|\widehat{Q}_1(s',\tilde{a}) - \widehat{Q}_2(s',\tilde{a})\Big| \\
&\leq \max_{s,a}\gamma \sum_{s'\in\mathcal{S}} \widehat{P}(s'|s,a) \max_{\tilde{s},\tilde{a}:(\tilde{s},\tilde{a})\in\mathcal{D}}\Big|\widehat{Q}_1(\tilde{s},\tilde{a}) - \widehat{Q}_2(\tilde{s},\tilde{a})\Big| \\
&= \max_{s,a}\gamma \sum_{s'\in\mathcal{S}} \widehat{P}(s'|s,a)\|\widehat{Q}_1 - \widehat{Q}_2\|_\infty \\
&= \gamma\|\widehat{Q}_1 - \widehat{Q}_2\|_\infty,
\end{aligned}
\tag{13}
$$

where the last equality is due to $\sum_{s'\in\mathcal{S}} \widehat{P}(s'|s,a) = 1$. $\qquad\square$

To demonstrate that the sampling-based update rule in Eq. (2) converges to $\widehat{Q}^*$, we utilize an auxiliary result from stochastic approximation [62, 63].

**Theorem A.2.** *The random process $\{\Delta_t\}$ taking values in $\mathbb{R}^n$ and defined as*

$$\Delta_{t+1}(x) = (1 - \alpha_t(x))\Delta_t(x) + \alpha_t(x)F_t(x) \tag{14}$$

*converges to zero with probability 1 under the following conditions:*

*(1) $0 \leq \alpha_t \leq 1$, $\sum_t \alpha_t(x) = \infty$ and $\sum_t \alpha_t^2(x) < \infty$;*

*(2) $\|\mathbb{E}[F_t(x)|\mathcal{F}_t]\|_W \leq \gamma\|\Delta_t\|_W$, with $\gamma < 1$;*

*(3) $Var[F_t(x)|\mathcal{F}_t] \leq C(1 + \|\Delta_t\|_W^2)$, for $C > 0$.*

$W$ is a norm. In our proof it is supremum norm.

*Proof.* See [62, 63]. $\qquad\square$

**Lemma A.3.** *Given any initial estimation $\widehat{Q}_0$, the following update rule:*

$$\widehat{Q}_{t+1}(s_t, a_t) = \widehat{Q}_t(s_t, a_t) + \alpha_t(x_t, a_t)[r_t + \gamma \max_{a:(s_{t+1},a)\in\mathcal{D}} \widehat{Q}_t(s_{t+1}, a) - \widehat{Q}_t(s_t, a_t)], \tag{15}$$

*converges w.p.1 to the optimal action-value function $\widehat{Q}^*$ if*

$$0 \leq \alpha_t(s,a) \leq 1, \quad \sum_t \alpha_t(s,a) = \infty \quad and \quad \sum_t \alpha_t^2(s,a) < \infty,$$

*for all $(s,a) \in \mathcal{S} \times \mathcal{A}$.*

*Proof.* Applying Theorem A.2, we can demonstrate the convergence of the update rule in Eq. (15). Rewriting Eq. (15), we get

$$\widehat{Q}_{t+1}(s_t, a_t) = (1 - \alpha_t(s_t, a_t))\widehat{Q}_t(s_t, a_t) + \alpha_t(x_t, a_t)[r_t + \gamma \max_{a:(s_{t+1},a)\in\mathcal{D}} \widehat{Q}_t(s_{t+1}, a)] \quad (16)$$

Subtracting $\widehat{Q}^*(s_t, a_t)$ from both sides yields:

$$\widehat{Q}_{t+1}(s_t, a_t) - \widehat{Q}^*(s_t, a_t)$$
$$= (1 - \alpha_t(s_t, a_t))(\widehat{Q}_t(s_t, a_t) - \widehat{Q}^*(s_t, a_t)) + \alpha_t(x_t, a_t)[r_t + \gamma \max_{a:(s_{t+1},a)\in\mathcal{D}} \widehat{Q}_t(s_{t+1}, a) - \widehat{Q}^*(s_t, a_t)]$$
$$(17)$$

Let's define

$$\Delta_t(s, a) = \widehat{Q}(s, a) - \widehat{Q}^*(s, a) \quad (18)$$

and

$$F_t(s, a) = r + \gamma \max_{a':(s',a')\in\mathcal{D}} \widehat{Q}_t(s', a') - \widehat{Q}^*(s, a). \quad (19)$$

This results in the same random process as shown in Theorem A.2 Eq. (14). Therefore, proving $\lim_{t\to\infty} \widehat{Q}_t = \widehat{Q}^*$ is equivalent to demonstrating that $\Delta_t(s, a)$ converges to zero with probability 1. We only need to verify that the assumptions in Theorem A.2 are satisfied under the definitions of Eqs. (18) and (19).

The first assumption of Theorem A.2 aligns with the condition in Lemma A.3. This can be easily achieved by choosing $\alpha_t(s, a) = 1/t$, for instance.

For the second assumption of Theorem A.2, we have

$$\mathbb{E}[F_t(s, a)|\mathcal{F}_t] = \sum_{s'\in\mathcal{S}} \widehat{P}(s'|s, a)[r + \gamma \max_{a':(s',a')\in\mathcal{D}} \widehat{Q}_t(s', a') - \widehat{Q}^*(s, a)]$$
$$= (\widehat{\mathcal{B}}\widehat{Q}_t)(s, a) - \widehat{Q}^*(s, a) \quad (20)$$
$$= (\widehat{\mathcal{B}}\widehat{Q}_t)(s, a) - (\widehat{\mathcal{B}}\widehat{Q}^*)(s, a)$$

Thus,

$$\|\mathbb{E}[F_t(s, a)|\mathcal{F}_t]\|_\infty = \|(\widehat{\mathcal{B}}\widehat{Q}_t) - (\widehat{\mathcal{B}}\widehat{Q}^*)\|_\infty$$
$$\leq \gamma\|\widehat{Q}_t - \widehat{Q}^*\|_\infty \quad (21)$$
$$= \gamma\|\Delta_t\|_\infty,$$

with $\gamma < 1$.

For the third assumption of Theorem A.2, we obtain

$$Var[F_t(s)|\mathcal{F}_t] = \mathbb{E}[F_t(s) - \mathbb{E}[F_t(s)|\mathcal{F}_t]|\mathcal{F}_t]^2$$
$$= \mathbb{E}[F_t(s) - ((\widehat{\mathcal{B}}\widehat{Q}_t)(s, a) - (\widehat{\mathcal{B}}\widehat{Q}^*)(s, a))]^2$$
$$= \mathbb{E}[r + \gamma \max_{a':(s',a')\in\mathcal{D}} \widehat{Q}_t(s', a') - \widehat{Q}^*(s, a) - ((\widehat{\mathcal{B}}\widehat{Q}_t)(s, a) - (\widehat{\mathcal{B}}\widehat{Q}^*)(s, a))]^2$$
$$= \mathbb{E}[r + \gamma \max_{a':(s',a')\in\mathcal{D}} \widehat{Q}_t(s', a') - (\widehat{\mathcal{B}}\widehat{Q}_t)(s, a)]^2$$
$$= Var[r + \gamma \max_{a':(s',a')\in\mathcal{D}} \widehat{Q}_t(s', a')|\mathcal{F}_t]$$
$$(22)$$

To align with the RHS in the third assumption of Theorem A.2, we add and subtract a $\gamma \max_{a':(s',a')\in\mathcal{D}} \widehat{Q}^*(s', a')$ term:

$$Var[r + \gamma \max_{a':(s',a')\in\mathcal{D}} \widehat{Q}^*(s', a') + \gamma \max_{a':(s',a')\in\mathcal{D}} \widehat{Q}_t(s', a') - \gamma \max_{a':(s',a')\in\mathcal{D}} \widehat{Q}^*(s', a')|\mathcal{F}_t] \quad (23)$$

Given that $r$ is bounded, it follows that $r + \gamma \max_{a':(s',a')\in\mathcal{D}} \widehat{Q}^*(s', a')$ is also bounded. Moreover, the second part $\max_{a':(s',a')\in\mathcal{D}} \widehat{Q}_t(s', a') - \max_{a':(s',a')\in\mathcal{D}} \widehat{Q}^*(s', a')$ can be bounded by $\|\Delta_t\|_\infty$ with some constant. Hence, we obtain

$$Var[F_t(s)|\mathcal{F}_t] \leq C(1 + \|\Delta_t\|_\infty^2), \quad (24)$$

for some constant $C > 0$ under the supremum norm. Therefore, by Theorem A.2, $\Delta_t$ converges to zero with probability 1, implying that $\widehat{Q}_t$ converges to $\widehat{Q}^*$ with probability 1.

Finally, when we derive the greedy policy from $\widehat{Q}^*$ by selecting the action with the highest value at each state $\widehat{\pi}^* = \arg\max_a \widehat{Q}^*(s, a)$, $\widehat{\pi}^*$ is optimal for the empirical MDP $\widehat{\mathcal{M}}$. If the optimal trajectory for $\mathcal{M}$ is included in $\widehat{\mathcal{M}}$, then this trajectory is optimal for $\widehat{\mathcal{M}}$, i.e. $\widehat{\pi}^*$ will select actions following that trajectory. Therefore, in this case, the greedy policy derived from $\widehat{Q}^*$ is also optimal for $\mathcal{M}$. □

**Proposition 3.4.** *Assume $\mathcal{M}$ has deterministic transitions. If $\{\mathcal{D}_i\}$ are datasets collected from $\mathcal{M}$ with $\mathcal{D}_1 \subset \mathcal{D}_2 \subset \cdots \subset \mathcal{D}_n$ and corresponding empirical MDPs $\{\widehat{\mathcal{M}}_i\}$, then $\widehat{Q}_1^* \leq \widehat{Q}_2^* \leq \cdots \leq \widehat{Q}_n^*$.*

*Proof.* Consider two datasets $\mathcal{D}_i \subset \mathcal{D}_j$ and their corresponding empirical MDPs $\widehat{\mathcal{M}}_i, \widehat{\mathcal{M}}_j$. Let $\widehat{\mathcal{B}}_i$ and $\widehat{\mathcal{B}}_j$ be the corresponding empirical Bellman operators. We have:

$$
\begin{aligned}
(\widehat{\mathcal{B}}_i \widehat{Q}_i^*)(s, a) &= \sum_{s' \in \mathcal{S}} P(s'|s, a)[r + \gamma \max_{a':(s',a') \in \mathcal{D}_i} \widehat{Q}_i^*(s', a')], \\
(\widehat{\mathcal{B}}_j \widehat{Q}_j^*)(s, a) &= \sum_{s' \in \mathcal{S}} P(s'|s, a)[r + \gamma \max_{a':(s',a') \in \mathcal{D}_j} \widehat{Q}_j^*(s', a')].
\end{aligned}
\tag{25}
$$

Where the transition probabilities $P(s'|s, a)$ are the same for both $\widehat{\mathcal{B}}_i$ and $\widehat{\mathcal{B}}_j$. Then, we can write:

$$
\begin{aligned}
\widehat{Q}_i^*(s, a) - \widehat{Q}_j^*(s, a) &= \widehat{\mathcal{B}}_i \widehat{Q}_i^*(s, a) - \widehat{\mathcal{B}}_j \widehat{Q}_j^*(s, a) \\
&= \sum_{s' \in \mathcal{S}} P(s'|s, a)[r + \gamma \max_{a'_i:(s',a'_i) \in \mathcal{D}_i} \widehat{Q}_i^*(s', a'_i)] - \sum_{s' \in \mathcal{S}} P(s'|s, a)[r + \gamma \max_{a'_j:(s',a'_j) \in \mathcal{D}_j} \widehat{Q}_j^*(s', a'_j)] \\
&= \sum_{s' \in \mathcal{S}} P(s'|s, a)\gamma[\max_{a'_i:(s',a'_i) \in \mathcal{D}_i} \widehat{Q}_i^*(s', a'_i) - \max_{a'_j:(s',a'_j) \in \mathcal{D}_j} \widehat{Q}_j^*(s', a'_j)] \\
&\leq \sum_{s' \in \mathcal{S}} P(s'|s, a)\gamma[\max_{a'_i:(s',a'_i) \in \mathcal{D}_j} \widehat{Q}_i^*(s', a'_i) - \max_{a'_j:(s',a'_j) \in \mathcal{D}_j} \widehat{Q}_j^*(s', a'_j)] \\
&\leq \sum_{s' \in \mathcal{S}} P(s'|s, a)\gamma \max_{\tilde{a}:(s',\tilde{a}) \in \mathcal{D}_j} (\widehat{Q}_i^*(s', \tilde{a}) - \widehat{Q}_j^*(s', \tilde{a})) \\
&\leq \gamma \sum_{s' \in \mathcal{S}} P(s'|s, a) \max_{(\tilde{s},\tilde{a}) \in \mathcal{D}_j} (\widehat{Q}_i^*(\tilde{s}, \tilde{a}) - \widehat{Q}_j^*(\tilde{s}, \tilde{a})) \\
&= \gamma \max_{(\tilde{s},\tilde{a}) \in \mathcal{D}_j} (\widehat{Q}_i^*(\tilde{s}, \tilde{a}) - \widehat{Q}_j^*(\tilde{s}, \tilde{a})) = \gamma \max_{(s,a) \in \mathcal{D}_j} (\widehat{Q}_i^*(s, a) - \widehat{Q}_j^*(s, a)).
\end{aligned}
\tag{26}
$$

By repeatedly applying the inequality, we get

$$
\begin{aligned}
\widehat{Q}_i^*(s, a) - \widehat{Q}_j^*(s, a) &\leq \gamma \max_{(s,a) \in \mathcal{D}_j} (\widehat{Q}_i^*(s, a) - \widehat{Q}_j^*(s, a)) \\
&\leq \gamma \max_{(s,a) \in \mathcal{D}_j} (\gamma \max_{(s,a) \in \mathcal{D}_j} (\widehat{Q}_i^*(s, a) - \widehat{Q}_j^*(s, a))) = \gamma^2 \max_{(s,a) \in \mathcal{D}_j} (\widehat{Q}_i^*(s, a) - \widehat{Q}_j^*(s, a)) \\
&\leq \cdots \\
&\leq \gamma^n \max_{(s,a) \in \mathcal{D}_j} (\widehat{Q}_i^*(s, a) - \widehat{Q}_j^*(s, a)).
\end{aligned}
\tag{27}
$$

Taking the limit on both sides and considering that $0 < \gamma < 1$, we obtain $\widehat{Q}_i^*(s, a) - \widehat{Q}_j^*(s, a) \leq 0$, i.e., $\widehat{Q}_i^*(s, a) \leq \widehat{Q}_j^*(s, a)$.

Then for $\mathcal{D}_1 \subset \mathcal{D}_2 \subset \cdots \subset \mathcal{D}_n$ and their corresponding empirical MDPs $\{\widehat{\mathcal{M}}_i\}$, we have $\widehat{Q}_1^* \leq \widehat{Q}_2^* \leq \cdots \leq \widehat{Q}_n^*$. □

# B Experimental Details

## B.1 Cliffwalk

**Environment Details.** Cliffwalk is a simple navigation task introduced by [26] and is depicted in Fig. 8. The agent's objective is to navigate from the bottom-left state to the goal state (G) located at the bottom right. The environment consists of 48 states, represented as two-dimensional coordinate axes $x$ and $y$. In our experiments, we present it as a $4 \times 12 \times 3$ binary array. The dimensions 4 x 12 correspond to the grid's shape, while the 3 denotes three channels representing specific game objects. The first channel shows the goal's position, the second channel shows the cliff's position, the last channel shows the agent's position. The action space comprises 4 directions: left, right, up, and down. The reward system is as follows: reaching the goal yields +1, falling into the cliff results in -1, and all other actions result in 0. We set the discount factor to 0.99 and the maximum episode steps to 100, as per [49].

Actions:

Figure 8: Illustration of the CliffWalk environment. Each grid represents a state, and the arrow indicates the optimal path from the start state (S) to the goal state (G).

The advantage of using a simple example like this is that we can pre-compute the optimal value and policy for a better understanding. Fig. 9 displays the optimal value for each state-action pair. Each grid contains four numbers in different directions, corresponding to the action values for the four actions: left, right, up, and down. The values, rounded to three decimal places, are highlighted in orange if they are optimal.

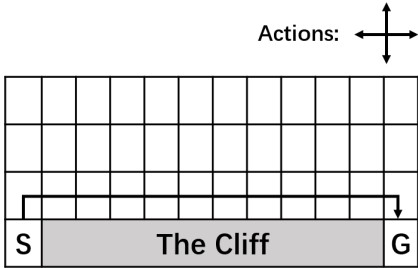

Figure 9: The optimal action value for each state-action pair. Each grid represents a state, with numbers in each direction indicating the action value for that state. The values in orange show the optimal actions.

**Implementation Details.** We train two DRL agents based on the Bellman update in Eq. (1) and the in-sample Bellman update in Eq. (2). We use a convolutional neural network that consists of a convolutional layer, followed by a fully connected layer. The convolutional layer has 16 $3 \times 3$ convolutions with stride 1, the fully connected layer has 128 units. The optimizer for the network is Adam [65] with a learning rate of 1e-4. We fill the replay memory with different transitions to simulate various scenarios. Detailed results are given in Appendix C.1.

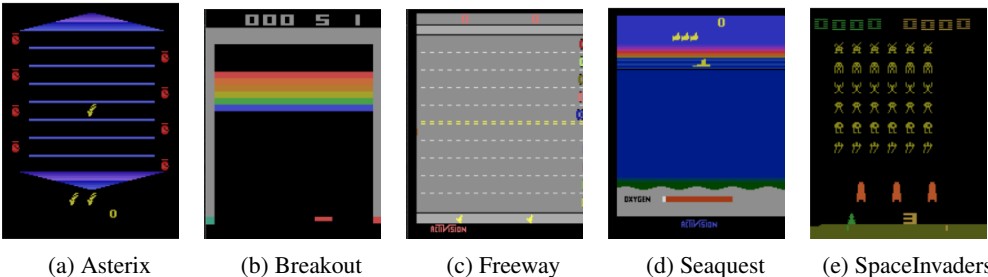

| (a) Asterix | (b) Breakout | (c) Freeway | (d) Seaquest | (e) SpaceInvaders |

Figure 10: Visualization of Atari environments.

## B.2 Arcade Learning Environment

**Environment Details.** The Arcade Learning Environment (ALE) [24] is a platform for evaluating the development of general, domain-independent AI technology. It provides an interface to a large number of Atari 2600 game environments with image input and discrete action space and with a variety of diverse mechanics. We show the visualization of some games in Fig. 10. Please refer to [24, 66] for more details.

**Implementation Details.** We use the same network for each algorithm as in the original paper. We search the learning rates for baselines among {1e-3,3e-4,1e-4} and report the best performance. For our method, we fix the learning rate to 1e-4 and search our regularization parameters $\alpha \in$ {0.05,0.1,0.5}. We run each experiment with 5 different random seeds. Each run consists of 10 million steps of interaction on Atari games and 2 million steps of interaction on MuJoCo tasks. The performance is evaluated by running 30 episodes after every 100k environmental steps. We proportionally reduce other parameters based on the interaction steps. The $\epsilon$-greedy exploration is linearly decayed from 1 to 0.01 in 1 million steps. The target network is updated every 1000 steps. The replay memory size is set as 100,000. The minibatch size is 32. The replay ratio is 0.25 [67], that is the $Q$ function is updated once per four environmental steps. The optimizer for the network is Adam. The discount factor is 0.99. Table 1 shows the details of hyper-parameters that used for all methods.

Table 1: Hyper-parameters of DQN on Atari environments.

| Hyperparameter | Value | Hyperparameter | Value |
|---|---|---|---|
| Batch size | 32 | Optimizer | Adam |
| Replay memory size | 100,000 | Initial exploration | 1 |
| Target network update frequency | 1,000 | Final exploration | 0.01 |
| Replay ratio | 0.25 | Exploration decay steps | 1M |
| Discount factor | 0.99 | Total steps in environment | 10M |

## B.3 MuJoCo

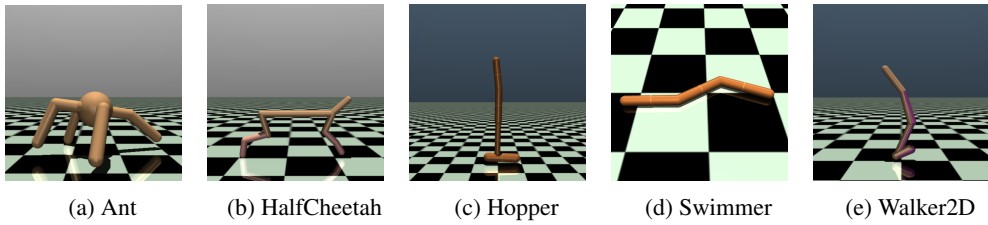

| (a) Ant | (b) HalfCheetah | (c) Hopper | (d) Swimmer | (e) Walker2D |

Figure 11: Visualization of MuJoCo environments.

**Environment Details.** MuJoCo [25] stands for Multi-Joint dynamics with Contact. It is a general purpose physics engine that aims to facilitate research and development in robotics. There are eleven MuJoCo environments in total, and we choose eight of them in our experiments: Ant, HalfCheetah, Hopper, Humanoid, InvertedDoublePendulum, Reacher, Swimmer and Walker2d. All of these environments are stochastic in terms of their initial state, with a Gaussian noise added to a fixed initial state in order to add stochasticity. The state spaces for MuJoCo environments in Gymnasium [35] consist of two parts that are flattened and concatenated together: a position of a body part or joint and its corresponding velocity. Often, some of the first positional elements are omitted from the state space since the reward is calculated based on their values, leaving it up to the algorithm to infer those hidden values indirectly. The action space is a vector that bounded within -1 and 1 for each dimension. Reward functions vary for each tasks. Please refer to [25, 35] for more details.

**Implementation Details.** For continuous control tasks, we compare our method with SAC [18, 39], TD3 [17], TRPO [40] and PPO [41]. SAC and TD3 are actor-critic methods that learn a $Q$ network and a policy network. TRPO and PPO are policy-based methods that only learn a policy network. We use the same network and parameters for each algorithm as in the original paper. Actor network has a tanh function before the final output, to control the action range at each dimension within [-1,1]. Our method is based on TD3, we maintain TD3's parameters the same as original paper [17] and search our particular parameter $\alpha \in \{0.05, 0.1, 0.5\}$. Other common parameters are shown in Table 2. We run each experiment with 5 different random seeds, and show the mean score and standard deviation with solid line and shaded area. Each run consists of 2 million steps of interaction on MuJoCo tasks. The performance is evaluated by running 10 episodes after every 10k environmental steps. The action is the output of mean value estimate without randomness.

For these baseline algorithms such as SAC[2], TD3[3] and XQL[4], we use their official code. We also refer some awesome public codebases such as RLzoo [68], Tianshou [69] and Dopamine [70]. We run all experiments on a server with Intel CPUs (Intel(R) Xeon(R) CPU E5-2650 v4 @ 2.20GHz) and NVIDIA GPUs (NVIDIA A5000). Each Atari experiment takes about one day, and each MuJoCo experiment takes about half a day.

Table 2: Hyper-parameters of TD3 on MuJoCo environments.

| Hyperparameter | Value | Hyperparameter | Value |
|---|---|---|---|
| Batch size | 256 | Target update rate ($\tau$) | 0.005 |
| Replay memory size | 1000,000 | Policy noise | 0.2 |
| Discount factor | 0.99 | Noise clip | 0.5 |
| Optimizer | Adam | Delayed policy update frequency | 2 |
| Learning rate | $3 \cdot 10^{-4}$ | Total steps in environment | 2M |

# C  Additional Experimental Results

## C.1  Toy Example on CliffWalk

We provide an in-depth discussion considering the replay memory containing different state-action pairs. We examine four cases:

(1) The replay memory only contains a failed trajectory and misses most of state-action pairs.

(2) The replay memory contains a sub-optimal trajectory and misses many state-action pairs.

(3) The replay memory contains optimal transitions but misses some sub-optimal state-action pairs.

(4) The replay memory contains optimal transitions and nearly all state-action pairs.

In each case, the replay memory is fixed. The first row in Fig. 12 shows the four cases. In this toy example with finite state-action space, we can track which actions are taken and which are not at

---

[2] https://github.com/haarnoja/sac
[3] https://github.com/sfujim/TD3
[4] https://github.com/Div99/XQL

a state. We sample batches of data and take the max over all actions to learn $Q$, and constrain the actions in the replay memory to learn $\widehat{Q}$.

The second row plot the error curves between $Q, \widehat{Q}$ and $Q^*, \widehat{Q}^*$. $\Delta(Q_1, Q_2)$ denotes the average absolute error on state-action pairs in the replay memory between $Q_1$ and $Q_2$:

$$\Delta(Q_1, Q_2) = \frac{1}{|\mathcal{D}|} \sum_{(s,a) \in \mathcal{D}} |Q_1(s,a) - Q_2(s,a)|. \tag{28}$$

We run five random seeds, the solid line shows the mean value and the shaded area shows standard error. It is clear to find that $\Delta(\widehat{Q}, \widehat{Q}^*)$ converges to 0, while $Q$ neither converges to $Q^*$ nor $\widehat{Q}^*$. In particular, when we have $Q^*(s,a) = \widehat{Q}^*(s,a) \ \forall (s,a) \in \mathcal{D}$. $\widehat{Q}$ converges to the global optimal while $Q$ still failed due to missing transitions. This highlights that the Bellman update is heavily affected by missing transitions, and the generalization of function approximation cannot correct it.

The last two rows present the final policies derived from $Q$ and $\widehat{Q}$ after learning. Red arrows represent actions that neither follow the optimal value $Q^*$ nor $\widehat{Q}^*$. The blue shading indicates the action values. We can observe that with function approximation, the policy based on the Bellman update cannot learn the (sub)optimal actions and overestimates action values. While the in-sample Bellman update converges to $\widehat{Q}^*$ and learn the best possible polices. When there are no successful samples in the first case, $\widehat{Q}$ still targets on current samples and converges to $\widehat{Q}^*$.

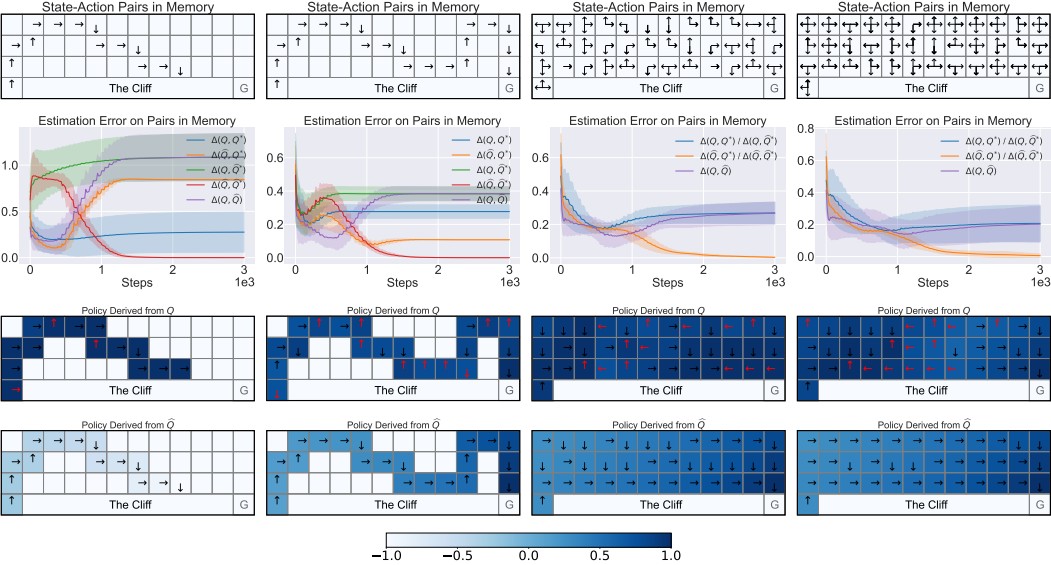

Figure 12: In the CliffWalk scenario, the rewards are as follows: +1 for reaching the goal, -1 for falling into the cliff, and 0 otherwise. The first row illustrates four cases with increasing transition coverage in the replay memory. The second row shows the errors during the learning process. The third and last rows present the final policies derived from $Q$ and $\widehat{Q}$. Red arrows represent incorrect actions that neither follow $Q^*$ nor $\widehat{Q}^*$. The blue shading indicates the action values.

## C.2 Overall Performance

We present the performance improvement curves for all environments when our method is applied to DQN and TD3 in Figs. 13 and 14. The solid line represents the mean score, and the shaded area indicates the standard deviation. $\widehat{Q}$ **online** is the IQL agent we run online that follows the learning rule Eq. (2) to learn $\widehat{Q}$. DQN and TD3 are agents that adhere to the learning rule Eq. (1) to learn $Q$. While $\widehat{Q}$ online aims to solve the empirical MDP, it lacks the exploration that expands the MDP. DQN and TD3 have exploration mechanism such as $\epsilon$-greedy and random noise, but they do not target the empirical MDP. Our method learn both $Q$ and $\widehat{Q}$, encompassing both steps of the Empirical MDP Iteration: solving the empirical MDP and augmenting the empirical MDP. Our method demonstrates significant improvement on all kinds of tasks compared with the methods that only learn $Q$ or $\widehat{Q}$. This suggests that the Empirical MDP Iteration is an effective enhancement for current DRL methods, and both steps are crucial for performance.

Next, we present the learning curves in comparison with other baselines in Figs. 15 and 16. The mean score and the standard error are represented by the solid line and shaded area, respectively. Our method either outperforms or is equivalent to (within a 10% difference) the best baselines on most tasks. Especially on MuJoCo tasks, our method outperforms other baselines by a large margin. These results validate our intuition and demonstrate that Empirical MDP Iteration is a superior approach than focusing on the original MDP from the beginning. It avoids bootstrapping error and thereby enhances sample efficiency.

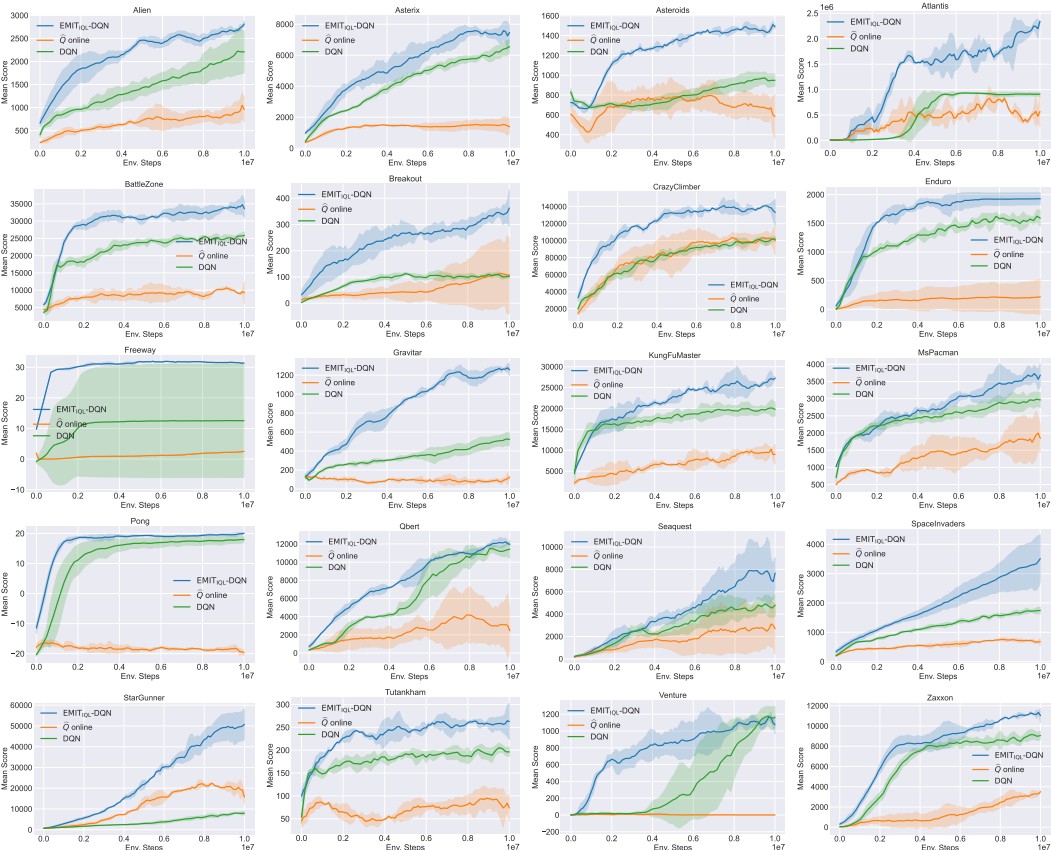

Figure 13: The performance improvement on Atari environments.

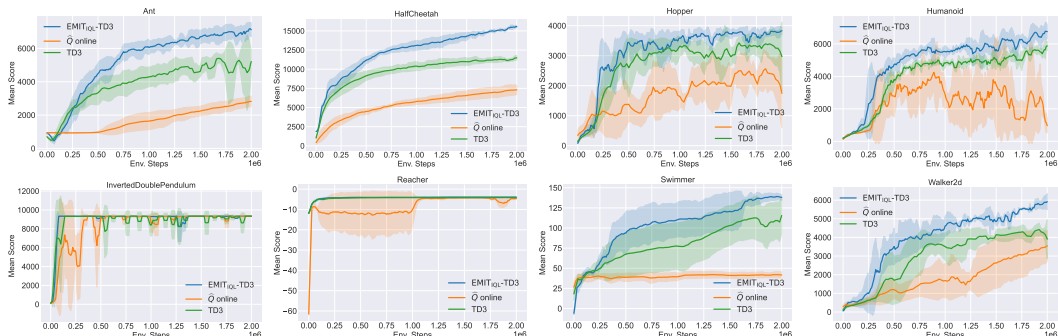

Figure 14: The performance improvement on MuJoCo tasks.

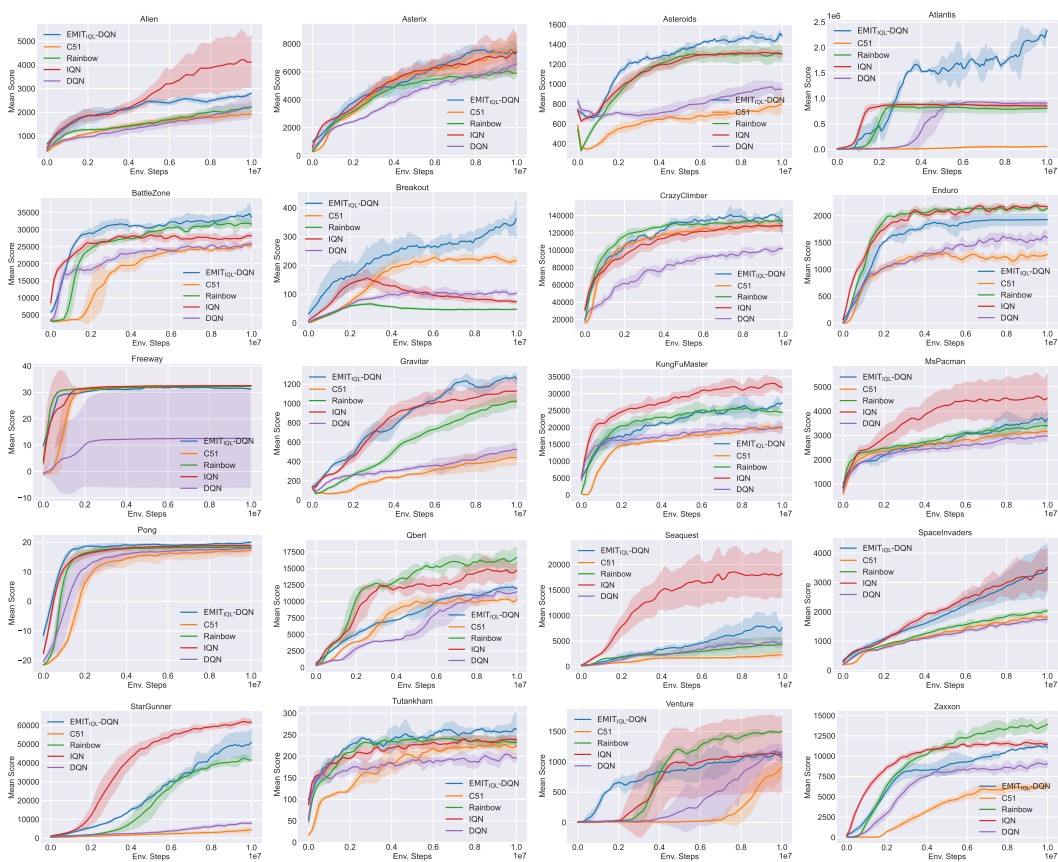

Figure 15: Overall performance on Atari environments. The learning curves show mean scores across 5 seeds and shaded area shows standard deviation.

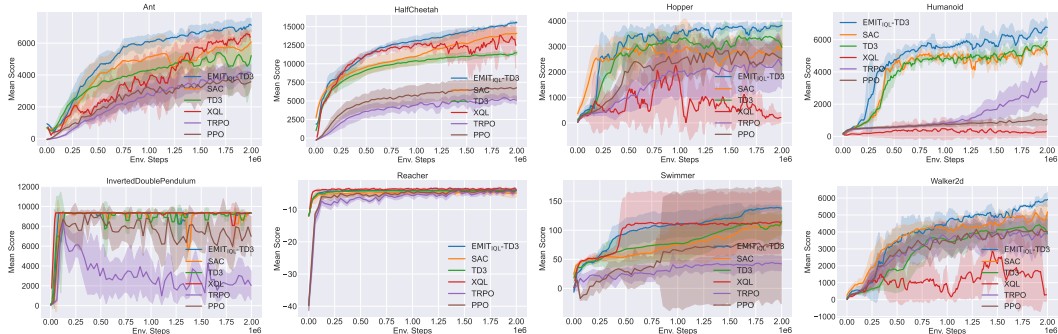

Figure 16: Overall performance on MuJoCo environments. The learning curves show mean scores across 5 seeds and shaded area shows standard deviation.

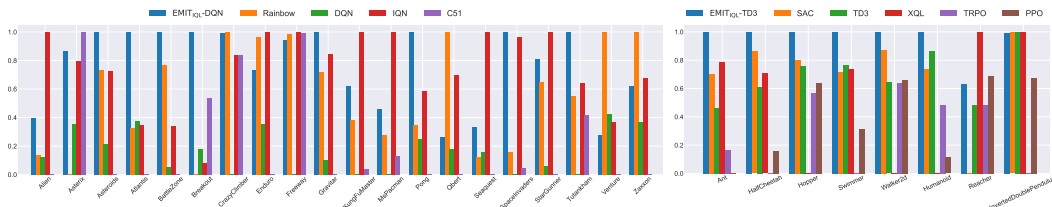

Figure 17: Performance on 20 Atari games and 8 MuJoCo tasks, with each method's score normalized between 0 and 1. EMIT outperforms or matches (within a 10% margin) the best baseline results in 19 of the 28 tasks.

## C.3 Analysis of Estimation Error

To determine whether the estimation or bootstrapping error is a ubiquitous issue on more complex environment, we run DQN and TD3 on Atari environments and MuJoCo tasks. These are based on the Bellman update in Eq. (1), and we denote the policy as $\pi$. We concurrently learn another IQL agent based on the in-sample Bellman update in Eq. (2), denoting the policy as $\hat{\pi}$. The policy $\pi$ interacts with the environment and collect data into a replay memory, from which both $\pi$ and $\hat{\pi}$ learn. As a result, both policies are learned with the identical dataset.

Figs. 18 and 19 shows the evaluation performance of $\pi$ and $\hat{\pi}$. We find that $\hat{\pi}$ performs similarly or even better than $\pi$ on most of the tasks. These results may seem counterintuitive, as active learning is commonly considered superior to passive learning [43]. This indicates that the in-sample Bellman update can learn a good policy from given data without active interaction with environments. On the contrary, the active policy $\pi$ does not fully utilize collected data and cannot find the best possible policy due to bootstrapping error. This highlights that bootstrapping from out-of-sample actions impedes sample-efficient learning and there should be room to improve it.

Next, we demonstrate that regularizing $Q$ with $\widehat{Q}$ significantly reduces the estimation error in $Q$. Estimation bias has long been a problem in value-based methods due to the combination of deep neural networks with the maximization operator in the Bellman optimality equation [20, 21]. Our method effectively eliminates overestimation or underestimation, achieving nearly accurate estimation throughout the learning process. This implies that there is almost no bootstrapping error, whether induced by the maximization operator or erroneous target estimation. At each evaluation, we obtain the true discounted Monte Carlo return $Q_{\text{True}}$ by running the current policy for 30 episodes and computing the actual discounted cumulative rewards at each state-action pair: $Q_{\text{True}}(s_t, a_t) = \sum_{t=t}^{T} \gamma^t r_t$. We then compute the averaged estimation difference between the learned $Q$ and ground truth $Q_{\text{True}}$, $V_{\text{diff}}(Q, Q_{\text{True}}) = \frac{1}{|(s,a)|} \sum_{(s,a) \in \mathcal{T}} (Q(s,a) - Q_{\text{True}}(s,a))$, where $\mathcal{T}$ is the set of evaluation trajectories, $|\mathcal{T}|$ denotes the total number of state-action pairs in $\mathcal{T}$. We compute $V_{\text{diff}}$ for our method and compare it with DQN and TD3.

Figs. 20 and 21 show that our method reduces the estimation error for most environments and keeps the difference near zero, indicating that the estimation is quite close to the true value and there is no serious overestimation or underestimation.

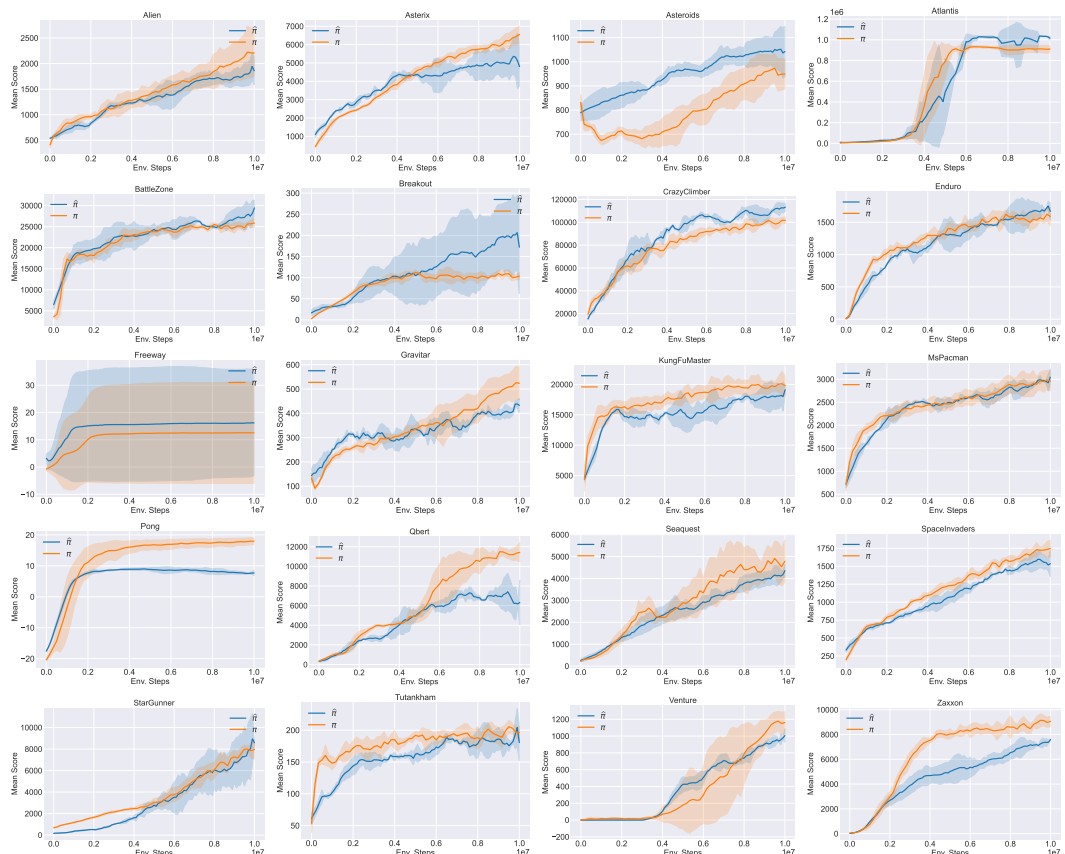

Figure 18: We run a DQN agent to interact with the environment, and concurrently learn action value networks $Q$ (DQN) and $\widehat{Q}$ (IQL) with the same data. The curves show the performance of argmax policies of $Q$ ($\pi$) and $\widehat{Q}$ ($\widehat{\pi}$).

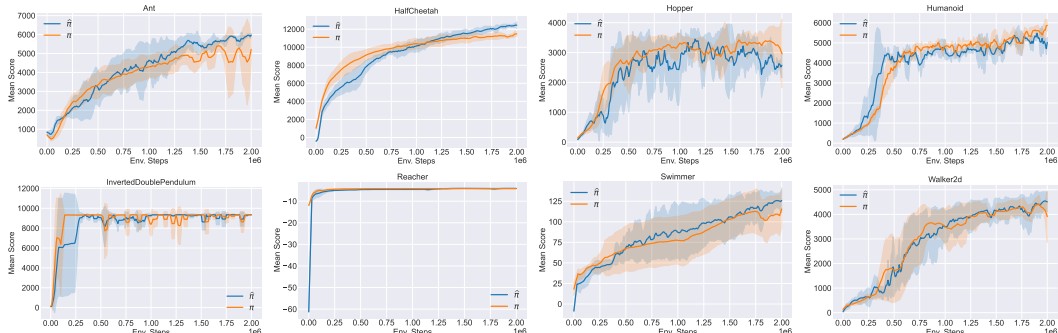

Figure 19: We run a TD3 agent to interact with the environment, and concurrently learn an IQL agent with the same data. The curves show the performance of TD3 ($\pi$) and IQL ($\widehat{\pi}$).

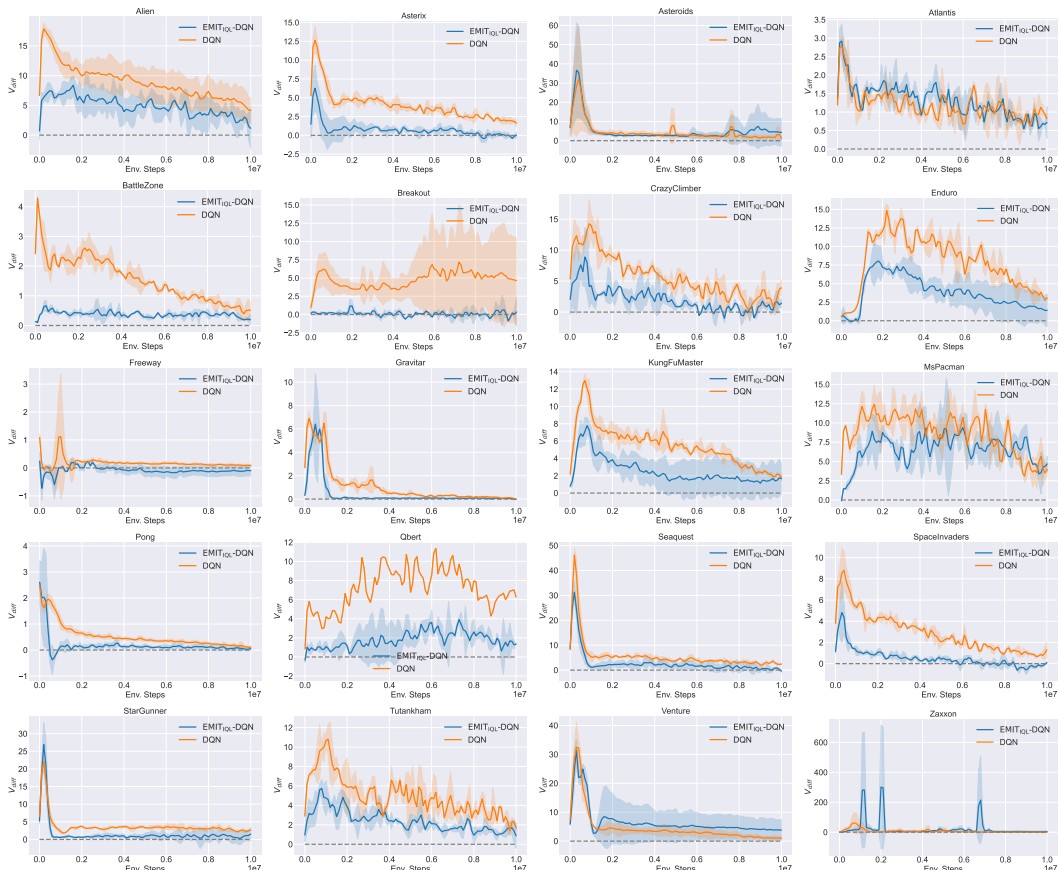

Figure 20: Estimation error on Atari environments. We compute the average difference between the estimated $Q$ value and the ground truth value during learning.

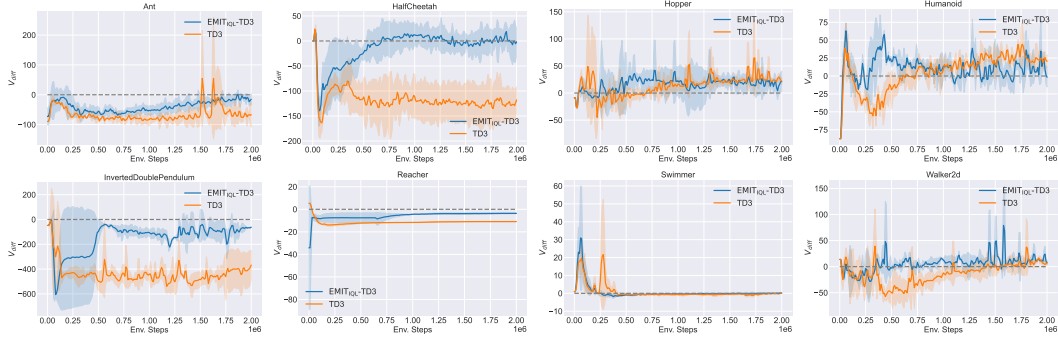

Figure 21: Estimation error on Mujoco environments. We compute the average difference between the estimated $Q$ value and the ground truth value during learning.

## C.4 Analysis of Policy Churn

Policy churn refers to the rapid change of the greedy policy in value-based reinforcement learning [34]. This phenomenon is primarily caused by deep learning with high-variance updates. Most of this policy change could be unnecessary, especially when learning converges. The per-state policy change between policies $\pi$ and $\pi'$ is defined as:

$$W(\pi, \pi'|s) := \frac{1}{2} \sum_a |\pi(a|s) - \pi'(a|s)|, \tag{29}$$

where $0 \leq W(\pi, \pi'|s) \leq 1$. When $\pi$ and $\pi'$ are greedy policies derived from action-value functions, $W(\pi, \pi'|s) = 1$ if $\pi$ and $\pi'$ choose different actions, otherwise $W(\pi, \pi'|s) = 0$. Thus it is simply a count of different actions after one step update. In agents that use a target network which is an older copy of the online network, it is easy to measure $W(\pi, \pi'|s)$ by comparing their argmax actions at the points in training where the target network lags behind by just one update.

When the target network lags behind by one update during the learning, for example at time step $t$, we sample a batch of data $\mathcal{D}$ and compute the average policy change $\overline{W}_t(\pi_{t-1}, \pi_t) = \frac{1}{|\mathcal{D}|} \sum_{s \in \mathcal{D}} W(\pi_{t-1}, \pi_t|s)$, where $|\mathcal{D}|$ is the size of the batch data. Then we compute the average policy change over the whole learning process as $\widetilde{W} = \frac{1}{|T|} \sum_{t \in T} \overline{W}_t$, where $T$ is the set of time steps that the target network lags behind by one update during the learning.

Fig. 22 shows the policy change on Atari environments. We can conclude that our method reduces policy churn compared to DQN. Policy churn is not entirely undesirable as it can also be a potentially beneficial form of implicit exploration. However, the frequency of change that benefits learning is still unclear and deserves further study.

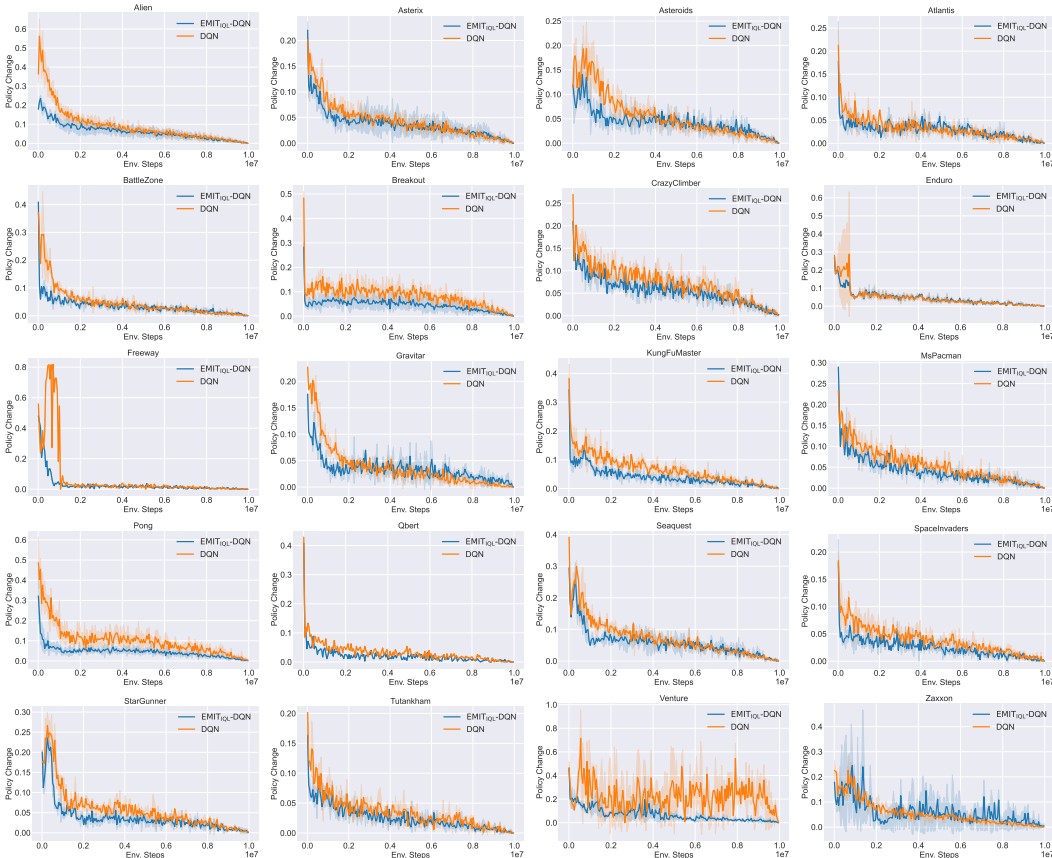

Figure 22: Policy churn curves on Atari environments. We compute the average policy change at each time step when the target network lags behind by one update.

## C.5 Ablation Study

Our method comprises two parts, the regularization term determined by parameter $\alpha$, and the exploration term designed based on $\delta(s, a)$. Figs. 23 and 24 illustrate the role of each part of our method. **w/o reg term** denotes that we set $\alpha = 0$. **w/o explore term** means we do not use our exploration mechanism. **w/o both** refers to the backbone methods without our two improvements. **full method** employs both improvements. We observe that both parts enhance the learning. The performance decreases more without regularization, indicating that following the empirical MDP iteration, the current exploration mechanisms such as $\epsilon$-greedy and random noise can make notable improvements. Adding an advanced exploration mechanism can further enhance the performance.

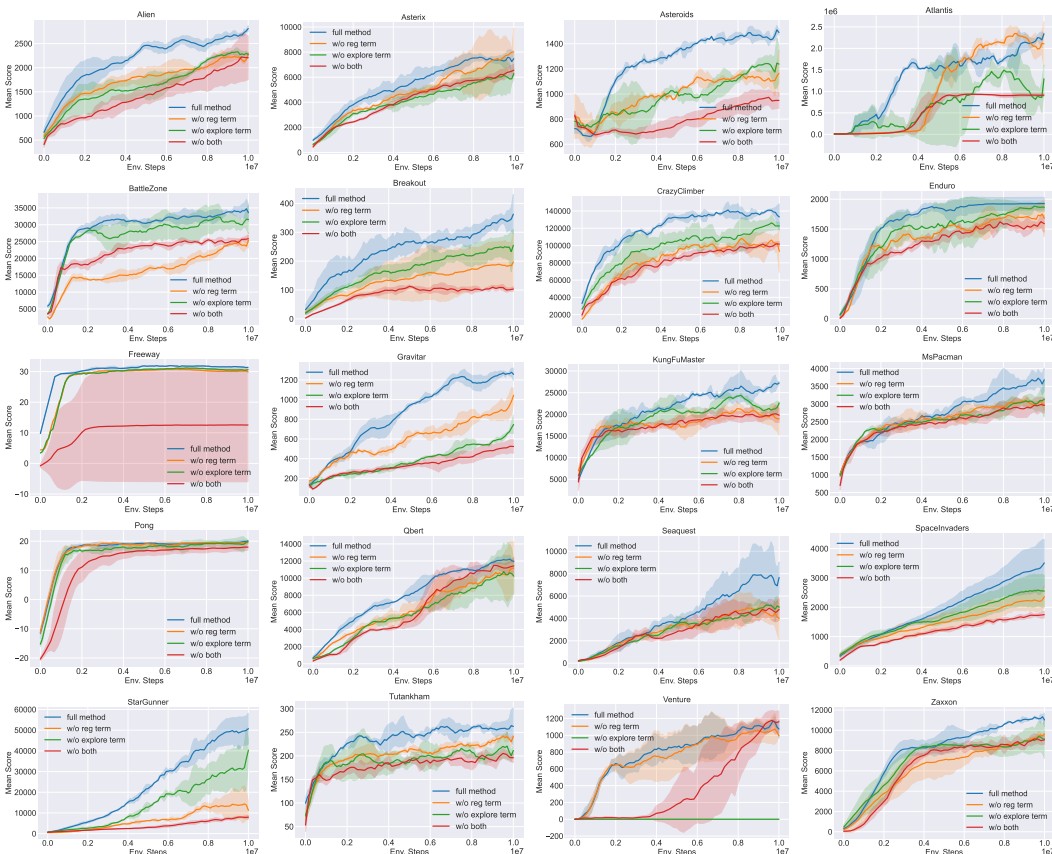

Figure 23: Ablations on Atari environments. Both the regularization and the exploration mechanism benefit the learning.

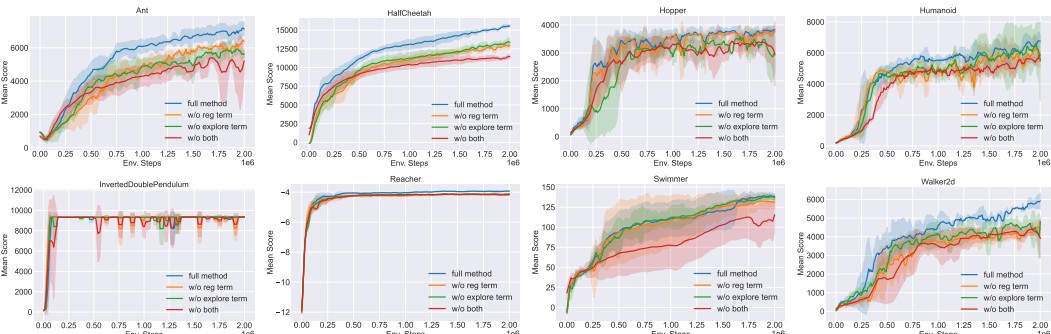

Figure 24: Ablations on MuJoCo environments. Both the regularization and the exploration mechanism benefit the learning.

