# OpenReview forum: "Exploiting the Replay Memory Before Exploring the Environment: Enhancing Reinforcement Learning Through Empirical MDP Iteration"
_NeurIPS.cc/2024/Conference — NeurIPS 2024 poster_

### Official Review · Reviewer_eYau · 2024-07-01

**Soundness:** 2
**Presentation:** 2
**Contribution:** 1
**Rating:** 5
**Confidence:** 5

**Summary:**

When using the Bellman update with incomplete data, the estimation error is hard to eliminate.
To solve this problem, the authors develop a novel framework EMIT, which can be used to enhance existing RL algorithms by iteratively solving a current empirical MDP for stable finite-time performance, and can progressively approach a solution to the original MDP.

**Strengths:**

1. The experiment is performed on both MuJoCo and Atari.
2.  Propositions 3.2-3.4 explain the estimation error in RL.

**Weaknesses:**

1. The running time of the different RL methods should be compared.
2. The proposed method should be compared with recent RL methods, such as TD7 and CrossQ [1, 2].
3. A head-to-head comparison between your memory buffer and the existing ones should be given [3].
4. From Algorithm 1, it is hard to see the difference between the proposed method and the existing ones.


If the authors can present their contributions clearly and open-source the code, I will  increase my rating.

[1]Fujimoto S, Chang W D, Smith E, et al. For sale: State-action representation learning for deep reinforcement learning[J]. Advances in Neural Information Processing Systems, 2024, 36.

[2] Bhatt A, Palenicek D, Belousov B, et al. CrossQ: Batch Normalization in Deep Reinforcement Learning for Greater Sample Efficiency and Simplicity[C]//The Twelfth International Conference on Learning Representations.

[3] Schaul T, Quan J, Antonoglou I, et al. Prioritized experience replay[J]. arXiv preprint arXiv:1511.05952, 2015.

**Questions:**

1. Can EMIT improve the performances of the recently proposed RL methods?
2. For continuous control, e.g., MuJoCo, in most cases, the current state-action pair will not be in the memory buffer D.
Thus, $R = -\infty$ for this state-action pair.
How to deal with this issue?

---

> ### Author Rebuttal · Authors · 2024-08-07
>
> - Can EMIT improve the performances of the recently proposed RL methods such as TD7 and CrossQ?
>
> We instantiate EMIT with TD7, and results are shown in Fig.1(b) in the rebuttal pdf. We find that EMIT can enhance the performance of TD7 on Ant and get similar performance on HalfCheetah. It may because TD7 is already a very strong baseline on HalfCheetah. We will discuss TD7 and CrossQ to provide a more comprehensive comparison in the revised manuscript.
>
>
> - For continuous control, e.g., MuJoCo, in most cases, the current state-action pair will not be in the memory buffer D. Thus, $R=-\infty$ for this state-action pair. How to deal with this issue?
>
> For the in-sample Bellman update as in eq.(3), we sample (s,a,r,s',a') in the buffer for the update. This r is not $-\infty$.
>
>
> - The running time of the different RL methods should be compared.
>
> We compare the running time of EMIT with DQN and TD3 using Frame Per Second (FPS) in the table below. EMIT consumes nearly double the time of DQN and TD3.
>
> || EMIT-DQN | Rainbow | DQN | IQN | C51 |
> |-| :-: | :-: |:-: |:-: |:-: |
> |FPS| 103.1 $\pm$ 4.2 | 124.0 $\pm$ 5.8 | 194.4 $\pm$ 5.7 | 179.1 $\pm$ 2.9 | 170.6 $\pm$ 5.6 |
>
> | | EMIT-TD3 | SAC | TD3 | XQL | TRPO | PPO |
> | :-: | :-: | :-: |:-: |:-: |:-: |:-: |
> | FPS | 42.2 $\pm$ 0.2 | 70.4 $\pm$ 0.3 | 79.4 $\pm$ 0.3| 68.2 $\pm$ 0.5 | 108.9 $\pm$ 3.1 | 166.6 $\pm$ 3.5 |
>
> Despite this limitation, as demonstrated in Figure 3 of our manuscript and summarized in the table below, EMIT still outperforms DQN and TD3 when using the same amount of computation, underscoring its efficacy.
>
> | | EMIT-DQN (5e6) | DQN (1e7) |
> | :-----| ----: | :----: |
> | Asteroids | 1338.9 $\pm$ 27.8 | 948.6 $\pm$ 61.9|
> | Atlantis | 1599569.9 $\pm$ 231875.6| 910471.2 $\pm$ 35727.5|
> | Breakout | 266.2 $\pm$ 40.4| 103.3 $\pm$ 8.0|
> | Gravitar | 949.7 $\pm$ 14.9| 523.5 $\pm$ 68.3|
>
> | | EMIT-TD3 (1e6) | TD3 (2e6) |
> | :-----| ----: | :----: |
> | Ant | 6059.5 $\pm$ 427.4| 5216.2 $\pm$ 720.1|
> | HalfCheetah | 13100.6 $\pm$ 639.5| 11481.2 $\pm$ 294.8|
> | Hopper | 3587.0 $\pm$ 225.0| 2968.1 $\pm$ 1145.1|
> | Humanoid | 5526.9 $\pm$ 378.3| 5876.5 $\pm$ 292.2|
>
> - A head-to-head comparison between your memory buffer and the existing ones should be given [3].
>
> Our first-in-first-out (FIFO) memory buffer is identical to the ones utilized in DQN and TD3. We have not incorporated any advanced memory buffer.
>
> - From Algorithm 1, it is hard to see the difference between the proposed method and the existing ones.
>
> Thank you for pointing this out. We will provide a more detailed description in the revised manuscript.
>
> - code
>
> In compliance with the rebuttal guidelines, which should not contain any links, we sent an anonymized link to the AC in a separate comment. We will open-source the code after the review process.

---

> > ### Comment · Reviewer_eYau · 2024-08-09
> >
> > I thank the authors for their detailed responses. I've increased my score.

---

> > > ### Author Response · Authors · 2024-08-09
> > >
> > > Thank you for taking the time to review our responses and for increasing your score. We greatly appreciate your feedback and support.

---

### Official Review · Reviewer_4uKn · 2024-07-08

**Soundness:** 3
**Presentation:** 3
**Contribution:** 3
**Rating:** 7
**Confidence:** 4

**Summary:**

The paper introduces the Empirical MDP Iteration (EMIT) framework, which enhances online reinforcement learning by regularizing algorithms with a sequence of empirical MDPs derived from replay memory data. By focusing on in-sample bootstrapping, EMIT ensures stable and unique convergence of Q-functions, leading to monotonic policy improvement (shown for deterministic MDPs). EMIT can be integrated with existing RL algorithms, effectively acting as a regularizer. Experiments with DQN and TD3 on Atari and MuJoCo benchmarks demonstrate that EMIT significantly reduces estimation errors and improves performance.

**Strengths:**

The paper is generally easy to follow and clearly written.

The proposed approaches appear to be simple and relatively original.

Good analysis and numerical results.

**Weaknesses:**

Reproducibility: The code was not provided in the supplemental material.
Representation: The text in Figure 2(a) is very small and unreadable.

**Questions:**

Can you plot $\Delta (Q, \hat Q) $ to see how far is Q from the empirical $\hat Q$?

In the discussion on page 4, in particular, when you state that "Q neither converges to Q* nor $\hat Q^*$, does this depend on the initialization of Q? How did you initialize Q?

**Limitations:**

Added computational complexity from learning two Q-functions

---

> ### Author Rebuttal · Authors · 2024-08-07
>
> - Can you plot to see how far is Q from the empirical Q?
>
> We plot the difference in Fig.4 in the rebuttal pdf. We find $\Delta(Q,\widehat Q)$ is similar to $\Delta(Q,\widehat Q^*)$ in the later stage of training since $\widehat Q$ will converge to $\widehat Q^*$.
>
> - when you state that "Q neither converges to $Q^*$ nor $\hat Q^*$, does this depend on the initialization of Q? How did you initialize Q?
>
> There is no dependency. In the extreme case where Q is initialized precisely to $Q^*$, then Q will converge to $Q^*$. Otherwise, Q has no guaranteed convergence to $Q^*$ whatever its initialization is. In our experiments, we initialize the Q as a randomly initialized neural network.
>
> - code
>
> In compliance with the rebuttal guidelines, which should not contain any links, we sent an anonymized link to the AC in a separate comment. We will open-source the code after the review process.
>
> - text in Figure 2(a) is very small and unreadable.
>
> Thank you for pointing this out. We will increase the font size in the revised manuscript.

---

> > ### Comment · Reviewer_4uKn · 2024-08-12
> >
> > Thank you for your response.

---

### Official Review · Reviewer_tCA6 · 2024-07-08

**Soundness:** 2
**Presentation:** 3
**Contribution:** 2
**Rating:** 5
**Confidence:** 3

**Summary:**

This paper introduces a novel framework called Empirical MDP Iteration (EMIT) to improve the stability and performance of reinforcement learning algorithms. Traditional reinforcement learning algorithms optimize a Markov Decision Process (MDP) using the Bellman equation, which can lead to unstable optimization when function approximation is used. And the EMIT framework addresses this by constructing a sequence of empirical MDPs from the replay memory and using an in-sample Bellman update to learn an estimated Q-function, denoted as $\widehat{Q}$. As claimed in the paper, this method restricts updates to in-sample data, ensuring convergence to a unique optimal $\widehat{Q}$ function and inducing monotonic policy improvement.

The paper demonstrates that EMIT can be integrated with existing reinforcement learning algorithms like DQN and TD3, acting as a regularizer to enhance their performance. The experimental results on Atari and MuJoCo benchmarks show that EMIT significantly reduces estimation errors and improves the performance of these algorithms. The authors provide theoretical analysis and extensive experiments to support their claims, highlighting the advantages of in-sample bootstrapping over traditional Bellman updates in reducing estimation errors and improving policy learning stability.

**Strengths:**

1. **Comprehensive Experimental Evaluation:** The paper presents extensive experimental results on both discrete action space environments (Atari) and continuous control tasks (MuJoCo). The diverse set of benchmarks provides a solid empirical foundation to validate the effectiveness of EMIT across different types of reinforcement learning tasks.

2. **Improved Stability and Performance:** The empirical results show that EMIT significantly enhances the stability and performance of existing reinforcement learning algorithms like DQN and TD3. By integrating EMIT as a regularizer, the paper demonstrates a clear reduction in estimation errors and notable policy improvements across various benchmarks.

**Weaknesses:**

**Limited Novelty in Core Ideas:** While the EMIT framework introduces in-sample bootstrapping for empirical MDPs, similar ideas have been explored in offline reinforcement learning and distributional reinforcement learning. The paper would benefit from a deeper differentiation from existing works, such as Implicit Q-Learning and In-Sample Actor-Critic. The authors should elaborate on how EMIT offers significant advancements over these methods beyond just integrating with online reinforcement learning algorithms.

I recommend the authors clearly delineate the unique contributions of EMIT compared to existing methods like Implicit Q-Learning and In-Sample Actor-Critic. Providing a comprehensive discussion on the theoretical and practical advancements offered by EMIT would strengthen the paper’s claim of novelty.

**Scalability Concerns:** The proposed framework requires maintaining and updating two Q-functions (Q and $\widehat{Q}$) simultaneously, potentially doubling the computational and memory requirements. This scalability issue is not sufficiently addressed in the paper. The authors should discuss how EMIT can be efficiently scaled to more complex environments or larger state-action spaces without significantly increasing the computational burden.

**Hyperparameter Sensitivity:** The paper lacks an in-depth analysis of the sensitivity of EMIT to its hyperparameters, particularly the regularization parameter $\alpha$ and the exploration bonus $\delta(s, a)$. Understanding the robustness of EMIT to different hyperparameter settings is crucial for its practical applicability. The authors should provide more insights or guidelines on how to select these hyperparameters in various scenarios.

**Questions:**

1. How does EMIT fundamentally differentiate itself from existing methods like Implicit Q-Learning and In-Sample Actor-Critic?
2. What are the limitations of the current exploration mechanism used in EMIT, and how does it compare to other advanced exploration strategies?
3. Can EMIT be effectively applied to real-world reinforcement learning problems, and what are the potential challenges in such applications?

I am willing to discuss and update my score until all the concerns are addressed.

**Limitations:**

The authors have discussed the limitations of their work, especially concerning the scalability and computational cost of maintaining and updating two Q-functions. However, the paper would benefit from a more detailed analysis of these limitations and their broader implications. Additionally, the potential societal impact of this research has not been thoroughly addressed.

---

> ### Author Rebuttal · Authors · 2024-08-07
>
> - Limited Novelty in Core Ideas. How does EMIT fundamentally differentiate itself from existing methods like Implicit Q-Learning and In-Sample Actor-Critic?
>
> The main contributions shown in our work is that in online RL, iteratively solving a sequence of empirical MDPs is better than just solving the original MDP, especially when the data is incomplete. We develop a novel framework EMIT to enhance existing online RL algorithms by iteratively solving current empirical MDPs. We showcase the effectiveness of our framework by using it to combine established methods for in-sample learning (i.e., Implicit Q-Learning, In-Sample Actor-Critic) and well-known online RL algorithms (i.e., DQN, TD3).
>
> - limitations of the current exploration mechanism used in EMIT, how does it compare to other advanced exploration strategies?
>
> The objective of exploration in EMIT is to grow the empirical MDP to better approximate the original MDP. Our current exploration mechanism employs the `principle of optimism in the face of uncertainty`, which has proven a useful heuristic in practice. However, its efficiency may wane when the environment is too hard to explore, or when estimated uncertainty is unreliable (e.g., in environments with a 'Noisy TV' problem [1]). In such instances, exploration methods like Go-Explore [2] or intrinsic motivation exploration [1,3,4] might be more suitable. We will discuss this in revised manuscript.
>
> [1] Burda, Yuri, Harrison Edwards, Amos Storkey, and Oleg Klimov. "Exploration by random network distillation." arXiv preprint arXiv:1810.12894 (2018).
>
> [2] Ecoffet, Adrien, Joost Huizinga, Joel Lehman, Kenneth O. Stanley, and Jeff Clune. "First return, then explore." Nature 590, no. 7847 (2021): 580-586.
>
> [3] Zhang, Tianjun, Huazhe Xu, Xiaolong Wang, Yi Wu, Kurt Keutzer, Joseph E. Gonzalez, and Yuandong Tian. "Bebold: Exploration beyond the boundary of explored regions." arXiv preprint arXiv:2012.08621 (2020).
>
> [4] Jarrett, Daniel, Corentin Tallec, Florent Altché, Thomas Mesnard, Rémi Munos, and Michal Valko. "Curiosity in Hindsight: Intrinsic Exploration in Stochastic Environments." arXiv preprint arXiv:2211.10515 (2022).
>
> - Can EMIT be effectively applied to real-world reinforcement learning problems, and what are the potential challenges in such applications?
>
> EMIT is designed to improve the performance of existing online RL algorithms. Thus, it should lead to improved performance on any `real-world` problems where existing online RL algorithms have shown to be effective. It is an interesting question on what kind of problems EMIT's advantage would disappear or unable to observe. We will try elaborate more on the limitations of EMIT in our revision. While EMIT could still encounter similar challenges as other RL algorithms, such as facing issues related to sample efficiency and safety, it is encouraging to note that EMIT not only boosts the sample efficiency of tested RL algorithms but also yields a more conservative estimate of the Q-function. This conservative estimation could potentially enhance the safety of the derived policy, positioning EMIT as a more preferable solution for real-world applications.
>
> - Scalability Concerns: The proposed framework requires maintaining and updating two Q-functions, potentially doubling the computational and memory requirements.
>
> We recognize the added computation cost in the "Limitations" section of our manuscript. However, the good news is that EMIT does not asymptotically make existing online RL algorithms more expensive over time. We thus believe EMIT enjoys similar `scalability` property as the online RL algorithms it intends to enhance.
> Future work to further improve EMIT's efficiency include designing an exploration strategy that is directly based on in-sample Bellman update, or improving runtime using two processes to update Q and $\hat Q$ in parallel. Despite this limitation, as demonstrated in Figure 3 of our manuscript and summarized in the table below, EMIT still outperforms DQN and TD3 when using the same amount of computation, underscoring its efficacy in the current format.
>
> | | EMIT-DQN (5e6) | DQN (1e7) |
> | :-----| ----: | :----: |
> | Asteroids | 1338.9 $\pm$ 27.8 | 948.6 $\pm$ 61.9|
> | Atlantis | 1599569.9 $\pm$ 231875.6| 910471.2 $\pm$ 35727.5|
> | Breakout | 266.2 $\pm$ 40.4| 103.3 $\pm$ 8.0|
> | Gravitar | 949.7 $\pm$ 14.9| 523.5 $\pm$ 68.3|
>
> | | EMIT-TD3 (1e6) | TD3 (2e6) |
> | :-----| ----: | :----: |
> | Ant | 6059.5 $\pm$ 427.4| 5216.2 $\pm$ 720.1|
> | HalfCheetah | 13100.6 $\pm$ 639.5| 11481.2 $\pm$ 294.8|
> | Hopper | 3587.0 $\pm$ 225.0| 2968.1 $\pm$ 1145.1|
> | Humanoid | 5526.9 $\pm$ 378.3| 5876.5 $\pm$ 292.2|
>
> - Hyperparameter Sensitivity
>
> We searched the best regularization parameter $\alpha$ in the range of [0.05, 0.1, 0.5] shown in Fig.3 in the rebuttal pdf. We find that a small value such as 0.05 or 0.1 suffices for the effective operation of EMIT. Regarding the exploration bonus, it does not introduce any hyperparameter. As demonstrated in the manuscript Fig.6(b), incorporating the exploration bonus notably enhances performance.

---

> > ### Comment · Reviewer_tCA6 · 2024-08-08
> >
> > Thanks for your rebuttal and I will maintain my score!

---

> > > ### Author Response · Authors · 2024-08-08
> > > **Follow-Up on Rebuttal and Feedback**
> > >
> > > Dear Reviewer,
> > >
> > > As you mentioned 'I am willing to discuss and update my score until all the concerns are addressed', we kindly ask if our responses have addressed your concerns. We appreciate your feedback and are happy to provide further clarification if needed.
> > >
> > > Many thanks,
> > > The Authors

---

### Official Review · Reviewer_jQwU · 2024-07-18

**Soundness:** 3
**Presentation:** 3
**Contribution:** 4
**Rating:** 7
**Confidence:** 3

**Summary:**

The authors transfer insights from IQL to online RL, demonstrating the performance of online RL can be enhanced by leveraging a Q-function that performs a max only over actions in the replay buffer when updating the Q-network. They use this network to:
* encourage exploration by driving the agent towards state-action pairs where the value predictions between a traditional Q-network and the IQL style Q-network differ
* they regularise the value predictions of a Q-network with a term equal to the difference between a traditional Q-network and an IQL style Q-network's value prediction

Overall, I like the paper and think it should be published.

**Strengths:**

* The authors demonstrate improved empirical performance on an impressive number of environments.

* They also perform ablation studies to analyse why their approach improves on previous methods (by considering how for example it reduces policy churn or how it improves the accuracy of Q-network value predictions).

* They provide theoretical results surrounding the convergence of EMIT and how it relates to Q-learning

**Weaknesses:**

* prior to reading this paper, I did not know how IQL worked. For example, it was initially confusing to me how an update like equation 2 could be done in continuous state spaces where you are unlikely to encounter the same state twice. I think a little bit more explanation of how this approach works (for example around equation 3) would help a reader like me that is not very familiar with the related work.
* it seems there are some insights that initially I thought we generated originally from your work but now I understand (e.g. Eq(2)) builds of prior work (IQL). Maybe that would be obvious to some but I think it would be good to make explicit your contributions compared to IQL etc
* Pretty hard for me to parse figure 4. I think something more suitable would show some summary statistics aggregated over all environments (with the bar charts in the appendix). Also why no error bars in figure 4?
* Would be interesting to see how it compares to basic regularization techniques (e.g. weight decay) and slightly more sophisticated methods for exploration (e.g. those that use intrinsic rewards). For example on venture it appears that the exploration term is contributing a lot to the improvement (possibly because it is a difficult environment in terms of exploration). I am not suggesting that EMIT must beat these methods, I just think it would add some context for the reader.

**Questions:**

Why integrate into TD3 instead of SAC when as far as I know SAC seems to outperform TD3?

**Limitations:**

The authors do discuss the limitations briefly in the conclusion, but I think having a larger limitation section where they are discussed more thoroughly would improve the paper.

---

> ### Author Rebuttal · Authors · 2024-08-07
>
> - Why integrate into TD3 instead of SAC when as far as I know SAC seems to outperform TD3?
>
> We integrate EMIT with DQN and TD3 mainly because they are directly built upon optimizing value-based Bellman equations, which aligns with our theoretical analysis and is exactly we aim to improve with EMIT. SAC, on the other hand, is originated from maximum entropy reinforcement learning with an entropy term. It is possible to implement EMIT with SAC. To show that, we've experimented EMIT with SAC, discovering that EMIT similarly enhances SAC's performance (Fig.1(a) in the rebuttal pdf).
>
> - a little bit more explanation of how IQL works
>
> We appreciate the suggestion and will incorporate a more detailed explanation of IQL in the revised manuscript.
>
> - it would be good to make explicit your contributions compared to IQL etc
>
> EMIT is a novel framework that iteratively solves a sequence of empirical MDPs, enhancing existing online RL algorithms. Other existing offline methods that solve empirical MDPs alternatively than IQL can also be adopted when instantiating EMIT. We will make this point more explicit in the revised manuscript.
>
> - summary statistics aggregated over all environments, and error bars in figure 4
>
> We summarize the aggregated results over all environments in Fig.2 in the rebuttal pdf. We will add error bars in fig.4 in the revised manuscript.
>
> - how it compares to basic regularization techniques (e.g. weight decay) and slightly more sophisticated methods for exploration (e.g. those that use intrinsic rewards).
>
> We appreciate the suggestion. In our current study, we concentrate on the theoretical analysis and empirical validation of the EMIT framework, employing a straightforward instantiation to showcase its efficacy. Indeed,  We aim to explore more sophisticated methods for exploration and regularization in our future research.
>
> - a larger limitation section
>
> We value your suggestion and will elaborate more on the limitations of this work.

---

### Author Rebuttal · Authors · 2024-08-07

We would like to thank all the reviewers for their comments, suggestions for improvement, and interest in the paper. Besides the detailed responses to each reviewer's comments, we submit a rebuttal pdf to supplement our responses. This pdf includes four figures to address the reviewers' concerns.
- Figure 1 shows the performance of EMIT with SAC and TD7, demonstrating that EMIT can also enhance the performance of other state-of-the-art RL algorithms.
- Figure 2 summarizes the aggregated results over all environments, providing a clearer comparison between EMIT and the baselines.
- Figure 3 shows the parameter sensitivity analysis of the regularization parameter $\alpha$, showing that a small value such as 0.05 or 0.1 suffices for the effective operation of EMIT.
- Figure 4 plots the difference between the Q-function and the empirical Q-function, showing that $\Delta(Q,\widehat Q)$ is similar to $\Delta(Q,\widehat Q^*)$ in the later stage of training since $\widehat Q$ will converge to $\widehat Q^*$.

---

### Decision · Program_Chairs · 2024-09-25

**Decision:**

Accept (poster)

**Comment:**

There is general agreement that the paper delivers a solid contribution to improving online RL performance. The reviewers have noted a number of presentation and clarification issues that the authors should address in the final paper.